**RESEARCH**

# Characterization of an eutherian gene cluster generated after transposon domestication identifies *Bex3* as relevant for advanced neurological functions

Enrique Navas-Pérez[1†], Cristina Vicente-García[2†], Serena Mirra[1,3,4,5†], Demian Burguera[1,6], Noèlia Fernàndez-Castillo[1,4,7], José Luis Ferrán[8], Macarena López-Mayorga[2], Marta Alaiz-Noya[9,10], Irene Suárez-Pereira[9,11], Ester Antón-Galindo[1], Fausto Ulloa[3,5], Carlos Herrera-Úbeda[1], Pol Cuscó[12,13], Rafael Falcón-Moya[9], Antonio Rodríguez-Moreno[9], Salvatore D'Aniello[14], Bru Cormand[1,4,7], Gemma Marfany[1,4,7], Eduardo Soriano[3,5,15], Ángel M. Carrión[9], Jaime J. Carvajal[2*] and Jordi Garcia-Fernàndez[1*]

\* Correspondence: j.carvajal@csic.es; jordigarcia@ub.edu
†Enrique Navas-Pérez, Cristina Vicente-García and Serena Mirra contributed equally to this work.
²Centro Andaluz de Biología del Desarrollo, CSIC-UPO-JA, Universidad Pablo de Olavide, 41013 Sevilla, Spain
¹Department of Genetics, Microbiology and Statistics, Faculty of Biology, and Institut de Biomedicina (IBUB), University of Barcelona, 08028 Barcelona, Spain
Full list of author information is available at the end of the article

## Abstract

**Background:** One of the most unusual sources of phylogenetically restricted genes is the molecular domestication of transposable elements into a host genome as functional genes. Although these kinds of events are sometimes at the core of key macroevolutionary changes, their origin and organismal function are generally poorly understood.

**Results:** Here, we identify several previously unreported transposable element domestication events in the human and mouse genomes. Among them, we find a remarkable molecular domestication that gave rise to a multigenic family in placental mammals, the *Bex/Tceal* gene cluster. These genes, which act as hub proteins within diverse signaling pathways, have been associated with neurological features of human patients carrying genomic microdeletions in chromosome X. The *Bex/Tceal* genes display neural-enriched patterns and are differentially expressed in human neurological disorders, such as autism and schizophrenia. Two different murine alleles of the cluster member *Bex3* display morphological and physiopathological brain modifications, such as reduced interneuron number and hippocampal electrophysiological imbalance, alterations that translate into distinct behavioral phenotypes.

**Conclusions:** We provide an in-depth understanding of the emergence of a gene cluster that originated by transposon domestication and gene duplication at the origin of placental mammals, an evolutionary process that transformed a non-functional transposon sequence into novel components of the eutherian genome. These genes were integrated into existing signaling pathways involved in the development, maintenance, and function of the CNS in eutherians. At least one of its members, *Bex3*, is relevant for higher brain functions in placental mammals and may be involved in human neurological disorders.

**Keywords:** Genetic novelty, Transposon domestication, *Bex3*, *Tceal*, Placental mammals, Gene cluster, Neurodevelopmental disorders, mTOR, Autism spectrum disorder

## Background

Newly evolved genes in a given lineage showing no homologs in other taxa are known as "orphan" or "taxonomically restricted" genes [1]. One of the most striking sources for the birth of lineage-restricted genes is the molecular domestication of transposable element (TE) proteins into novel coding genes [2], which are sometimes involved in the appearance of clade-specific traits and even true evolutionary novelties [3]. Despite their evolutionary relevance, a systematic search for domesticated transposons taking advantage of current and improved genomic annotations was lacking in human and mouse. We have identified now several domestication cases in these species. Among them, we highlight here a previously unreported event that took place at the origin of eutherian mammals, which gave rise to a multigenic family known as *Bex/Tceal* and shaped a cluster of 14 genes on the X chromosome of the placental ancestor. While most are scarcely studied, several reports have linked some of these genes to processes such as cancer proliferation [4–9], cellular reprogramming [10] and differentiation [11], or cell cycle modulation [12, 13]. To investigate the function of this gene family at the organism level, we generated and phenotyped mutant mice for one of its members, *Bex3*. Mutants showed molecular, cellular and anatomical alterations in the brain, as well as important neurological and behavioral alterations typical of neurodevelopmental defects. Altogether, we describe the evolutionary pathways of a TE-derived gene family that integrated into complex molecular routes, while underscoring its impact on neural development and its neuropsychiatric significance.

## Results

### Identification of genes derived from molecular domestication of TEs

In order to detect new transposon domestication events in the mammalian lineage, we looked for protein-coding genes made up by TE-derived sequences. By identifying genes with a coding sequence overlapping greater than 50% with annotated TEs and present in more than one species, we obtained a list of 28 and 9 candidates in the human and mouse genomes, respectively (Additional file 1: Table S1). We recovered well-known TE-derived genes, such as *syncytin-1* [14], *syncytin-2* [15], *SETMAR* [16], and the *ZBED* [17] and *PNMA* families [18]. More ancient domesticated transposons like the vertebrate-specific *RAG1* and *RAG2* genes [3] were not detected, probably due to higher sequence divergence in relation to the ancestral TE element. We also found new

putatively domesticated TEs, most with unknown function, confined to either primate or *Mus* species. However, our attention was drawn to the remarkable case of *Tceal7*, a gene present in all major groups of placental mammals. Due to its broad phylogenetic range, we decided to study this molecular domestication event in further detail.

### Molecular composition of the *Tceal7* gene

*Tceal7* is a small gene consisting of two 5′ non-coding exons and a third exon that includes the whole open reading frame (ORF) and the 3′UTR. We observed an overlap of 76% between the *Tceal7* ORF sequence and that of HAL1b, a non-LTR retrotransposon belonging to the long interspersed nuclear element-1 (LINE-1 or L1) superfamily (Fig. 1a). Moreover, the last 18 nucleotides of the *Tceal7* ORF and most of the 3′UTR originated from two other L1 subfamilies: the elements L1MEe and L1ME4a (Fig. 1a and Additional file 1: Fig. S1). We observed a partial retention of the original TE coding frame, as human TCEAL7 amino acid sequence shares 31.7% identity with the ORF1p of HAL1b (Fig. 1b). From these results, we determined that the *Tceal7* gene arose from a composite sequence derived from two L1 elements (Fig. 1).

Although *Tceal7* stands out as the most similar to the ancestral L1 retrotransposon sequences, it is but a representative member of a multigenic family called *Bex/Tceal*. This family forms a gene cluster on the X chromosome of placental mammals, with no detectable orthologs outside this clade [21]. In humans, the cluster consists of 5 *Bex* (brain-expressed X-linked) and 9 *Tceal* (transcription elongation factor A (SII)-like) genes, spanning ∼ 1.5 Mb. In mouse, the number is reduced to 11 genes due to the lineage-specific loss of *Tceal2*, *Tceal4*, and *Bex5*. Interestingly, we detected the presence of clustered *Bex/Tceal* genes in each major eutherian lineage (Additional file 1: Fig. S2), confirming that the origin and expansion of the family took place after the divergence of the marsupial-placental clades, and before the radiation of the latter. In agreement, HAL1b, L1MEe, and L1ME4a elements have been suggested to be active retrotransposons during the appearance of early eutherians ∼ 150 Mya [22].

### Evolutionary diversification of the *Bex/Tceal* family

BEX and TCEAL protein sequences are relatively divergent [21], likely due to low selective constraints after the retrotransposon domestication. However, both families share a conserved region toward their C-terminal end (Additional file 1: Fig. S3A). BEX proteins, with the exception of BEX4, are predicted to have a coiled coil within this region [23–25], and we found that TCEAL1, TCEAL7, and TCEAL8 are also predicted to harbor a C-terminal coiled coil domain (Additional file 1: Fig. S3A). Remarkably, most of the protein-protein interactions of BEX1, BEX2, and BEX3 have been mapped to this domain [24], highlighting its potential functional relevance. Furthermore, BEX proteins have been classified as intrinsically disordered proteins (IDPs) [24]. We found that TCEAL proteins, as previously described for BEX members [24], are predicted to have a disordered N-terminal region and C-terminal α-helices (Additional file 1: Fig. S4). Interestingly, we also detected all these structural features in the ancestral transposon HAL1b (Additional file 1: Fig. S4), suggesting that these features were preserved along the domestication process and inherited by the *Bex/Tceal* genes.

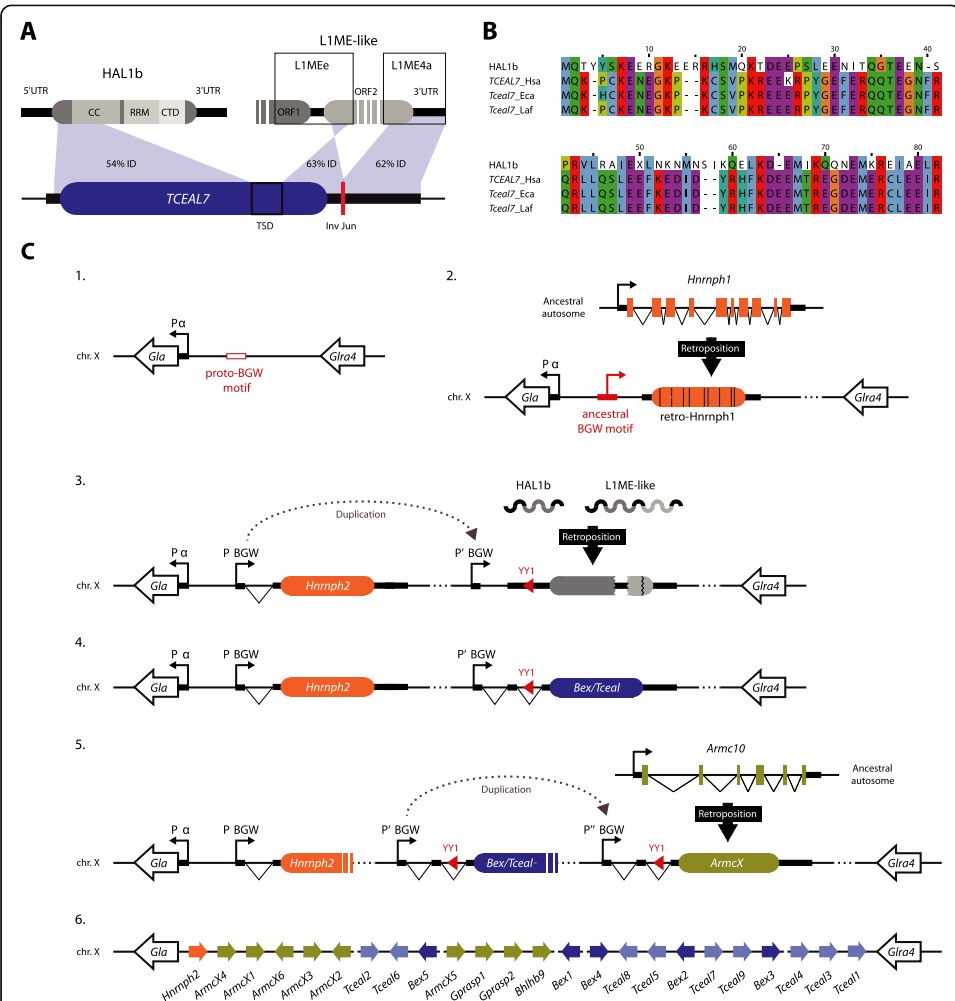

**Fig. 1** Transposon domestication and co-option of a DNA motif originated the *Bex/Tceal* gene family. **a** Diagram showing the domestication of L1 fragments to originate the *TCEAL7* gene ORF and 3′UTR. The target site duplication (TSD) and the inversion junction (Inv Jun) resulting from the transposition process [19] are shown. The identity (ID) value was obtained by aligning the transposon nucleotide sequences with the corresponding region of the human *TCEAL7* gene. L1MEe and L1ME4a-derived sequences of TCEAL7 are drawn as belonging to the same L1ME-like retrotransposon. The protein domains of the ORF1p of HAL1b are indicated: CC, coiled coil; RRM, RNA-recognition motif; and CTD, carboxy-terminal domain. **b** Protein alignment of the HAL1b ORF1p and the N-terminus of TCEAL7 proteins from three placental species: Hsa, *Homo sapiens*; Eca, *Equus caballus*; Laf, *Loxodonta africana*. **c** Diagram showing the proposed evolutionary scenario for the formation of the BGW cluster. (1) A proto-BGW motif lay upstream of the alpha-galactosidase (*Gla*) promoter (P α) in the X chromosome of the ancestor of eutherians and metatherians. (2) In the eutherian lineage, a *Hnrnph1* transcript was retrotranscribed and inserted upstream of *Gla* and next to the ancestral BGW motif. (3) The co-option of this BGW motif (P BGW) gave rise to *Hnrnph2* retrogene. This genomic region was duplicated, and retrotransposons HAL1b and L1ME-like inserted nearby. (4) The new retrotransposon-derived ORF fell under the transcriptional influence of a BGW motif (P′ BGW). The YY1 binding site derived from the 5′UTR of HAL1b [20] was preserved. (5) The BGW motif (P″ BGW) and the YY1 binding site of a *Bex/Tceal* gene duplicated upstream of a retrocopy of the *Armc10* gene, giving rise to the *ArmcX* ancestral gene. (6) Before the diversification of placentals, the *Bex/Tceal* and *ArmcX* gene families expanded forming the BGW cluster

Previous studies using a reduced number of mammalian species reported gene conversion events and positive selection signatures on some *Bex/Tceal* genes [21, 26]. We decided to perform an expanded analysis adding species from the major eutherian clades, and found homogenization of coding sequences among three

groups of *Bex/Tceal* paralogs (*Tceal2* with *Tceal4*; *Tceal3* with *Tceal5* and *Tceal6*; and *Bex1* with *Bex2*) across all studied lineages (Additional file 1: Fig. S3B). Moreover, after filtering out sequences experiencing gene conversion [27], we detected several sites with signatures of positive selection, mainly in the *Bex* subfamily tree, and an episode of positive selection after the branching of *Bex5* (Additional file 1: Fig. S5). This suggests that an ancestral gene within the branch leading to *Bex3* and *Bex4* went through an adaptive process in an eutherian ancestor before the diversification of placental mammals.

### Neighboring DNA sequence co-option as a central regulatory element

Within the same eutherian-specific chromosomal region containing *Bex* and *Tceal* genes, there are multiple retrocopy genes belonging to another gene family (the *Armcx* family, also known as *ALEX*) [28]. Despite no protein similarity between both families, they share a homologous DNA sequence motif in their promoter region known as the BGW motif [21], a ~ 60 base pair (bp)-long sequence containing an internal E-box, which has been shown to be essential for the regulation of mouse *Tceal7* [29] and human *ARMCX1* [30] expression. The promoter region of *Hnrnph2*, another eutherian-specific retrogene located at the centromeric end of the cluster, also harbors this unique motif [21]. This multiplicity of BGW motifs raises the question of how genes with three independent origins ended up with separate, but homologous, regulatory elements. The promoter of *Hnrnph2* is bidirectional and shared with the galactosidase-alpha (*GLA*) gene [31]. We found a BGW-like motif upstream of the *GLA* promoter in marsupials that lacks an eutherian-restricted 11 bp sequence required for the proper transcription of human *ARMCX1* gene [30] (Additional file 1: Fig. S6). Therefore, the origin of the BGW motif can be traced back to sequences already present in the *GLA* promoter of the last therian common ancestor. Although we cannot determine the precise order of the events leading to the assembly of the whole cluster, the inferred co-option of the BGW motif by the ancestors of the *Bex/Tceal* and *Armcx* families allows us to reconstruct the main steps of this evolutionary process (diagrams depicted in Fig. 1c, see legend for details).

### Brain expression of the *Bex/Tceal* genes and deregulation in neuropsychiatric disorders

Information about expression patterns for *Bex/Tceal* genes is disperse, with data restricted to human, mouse, or rat [12, 29, 32–37]. We gathered publicly available transcriptomic data from eight homologous adult organs of five species belonging to main placental lineages. We observed that most genes present a tissue-enriched pattern, with brain being the organ showing the highest expression levels for most paralogs across species (Fig. 2a). Although brain is an organ where many genes tend to be expressed [38], this result suggests that some of the neural functions reported in mouse and human for this gene family [11, 12, 39] might be conserved among the eutherian clade.

We also investigated the expression patterns of this group of genes during mouse development by in situ hybridization and observed a subset of *Bex/Tceal* genes being highly and widely expressed during mouse embryogenesis, especially *Bex* genes (Fig. 2b and Additional file 1: Fig. S7). *Bex3* expression was particularly strong in

the central nervous system during murine development compared to other members of the family (Fig. 2b).

By analyzing publicly available human transcriptomic data of autism spectrum disorder (ASD) and schizophrenia, two well-studied neuropsychiatric disorders, we found a significant decrease in the expression of *BEX/TCEAL* genes in patients compared to controls in different brain regions and datasets (Additional file 1: Table S2), being *BEX* but not *TCEAL* genes significantly enriched among the differentially expressed genes in most datasets (Additional file 1: Table S3). Notably, *BEX3* is located in the interval associated with the neurological features of patients diagnosed with early-onset neurological disease trait (EONDT), which harbor different genomic deletions encompassing *BEX/TCEAL* genes [40–42].

## Activation of neural tube progenitor proliferation by BEX/TCEAL proteins in a non-eutherian vertebrate

Next, we investigated the putative function of the inferred, original TE composite sequence that later evolved into the *Bex/Tceal* family. To understand the potential physiological response of this ancient element in the eutherian ancestor, we aimed to mimic the original scenario by using a non-eutherian organism. For this purpose, we synthetically reconstructed a version of the ancestral *Bex/Tceal* protogene, termed *HALEX* (**HAL**1b-derived on **e**utherian **X** chromosome), based on TE consensus sequences (see "Methods"). Moreover, we also studied two *Bex/Tceal* members: *Tceal7*, the gene with the highest sequence similarity to the original protogene; and *Bex3*,

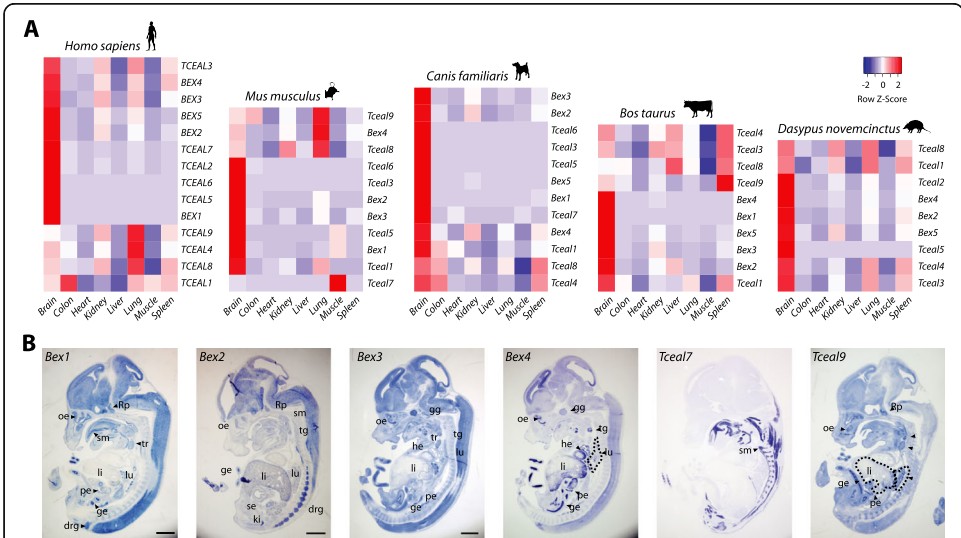

**Fig. 2** *Bex/Tceal* genes show a tissue-enriched expression pattern, particularly in neural organs. **a** Heatmaps showing relative gene expression levels of *Bex* and *Tceal* genes in eight adult tissues/organs using RNA-seq data from five eutherian species: human (*Homo sapiens*), mouse (*Mus musculus*), dog (*Canis familiaris*), cow (*Bos taurus*), and nine-banded armadillo (*Dasypus novemcinctus*). In the analyzed organisms, most genes are highly expressed in the brain, particularly *Bex* genes. Highest and lowest expression levels are represented in red and blue, respectively. **b** In situ hybridization of *Bex* and *Tceal* genes, which show high expression levels (*Bex1, Bex2, Bex3, Bex4, Tceal7,* and *Tceal9*) in E13.5 mouse embryos. Sagittal sections of the whole embryo are shown. drg, dorsal root ganglion; ge, gut epithelium; gg, gasserian ganglion; he, heart; ki, kidney; li, liver; lu, lung; oe, olfactory epithelium; pe, pancreatic epithelium; Rp, Rathke's pouch; se, stomach epithelium; sm, skeletal muscle; tg, thyroid gland; tr, thymic rudiment. Scale bar: 1 mm

which shows the strongest expression in the embryonic nervous system. We electroporated murine *Bex3*, *Tceal7*, and *HALEX* genes into the neural tube of chicken embryos at stage HH12 to investigate their capacity to elicit a cellular response in neuronal progenitors. Expression of *Bex3* and *Tceal7*, but not *HALEX*, generated a significant increase in cell proliferation in the chicken embryonic neural tube, similarly to previous reports in mammalian cultured cells [11, 43] (Additional file 1: Fig. S8). Although the heterologous approach we have used cannot reproduce the regulatory and signaling environments of the eutherian ancestor, the observation that only the murine constructs, but not the ancient element, produced a measurable response suggests that the current ability of *Bex/Tceal* genes to effectively modulate cellular physiology was not present in the ancestral protogene, being acquired during eutherian evolution.

### Generation of two independent *Bex3* mutant lines

While functional studies for the *Bex/Tceal* genes have focused mainly on in vitro assays, little is known about their role at the organism level. Based on previous reports linking *Bex3* to neuronal physiology [11, 43, 44], the analyses on patients with neurological features harboring deletions encompassing *BEX/TCEAL* genes [40–42], and its neural-enriched expression in adult and embryonic tissues, we decided to generate mouse mutant lines for this gene. We used CRISPR-Cas9 technology to generate mutant alleles for *Bex3* in mice and selected two for in-depth characterization (Fig. 3a and Additional file 1: Fig. S9). One of them, namely *Bex3$^{KO}$*, carried a 196-bp deletion that caused a frameshift mutation, introducing a premature stop codon that led to a truncated coding sequence. The second line, which we named *Bex3$^{\Delta(24-72)}$*, carried a 147-bp deletion that removed 49 amino acids of the central core of the protein, specifically the pro-apoptotic domain, and retained the C-terminal coiled coil domain required for BEX3 dimerization, nuclear import, ubiquitination [44], and interaction with its multiple partners [24]. Therefore, this mutation potentially expresses a protein lacking an essential functional domain, which is likely to act as a hypomorph or a dominant negative allele, as it has been shown to be the case in cell culture experiments [44].

### Anatomical alterations in skull and brain of *Bex3* mutant mice

Homozygous mutant mice for both lines could be distinguished from wild-type counterparts by external observation of subtle facial differences (Fig. 3b, c), hence skull measurements were taken out in order to identify the origin of these alterations. Snout-to-midbone length (D2 in Additional file 1: Fig. S10) tended to be smaller in the mutants, while the width of the eye sockets (D24 in Fig. 3c) was significantly larger in *Bex3$^{KO}$* mice. Further, the ratio between these measurements was reduced by 19.4% ± 3.4 in *Bex3$^{KO}$* ($p = 0.001$) and 10.2% ± 3.4 in *Bex3$^{\Delta(24-72)}$* ($p = 0.064$), suggesting abnormal frontal bone morphology that resulted in a rounder eye appearance. Additionally, lack of fully functional *Bex3* also led to increased size of posterior parameter measurements (D5, D19-D21 in Fig. 3c and Additional file 1: Fig. S10), indicating that the bone cavity harboring the cerebellum was larger in the mutants, particularly in *Bex3$^{KO}$* mice.

The overall brain and cerebellum anatomical structures of *Bex3$^{KO}$* and *Bex3$^{\Delta(24-72)}$* mice appeared close to normal, although they showed a reduction in cortical surface and size, as well as in cerebellum size (Fig. 3d, e) despite the layered

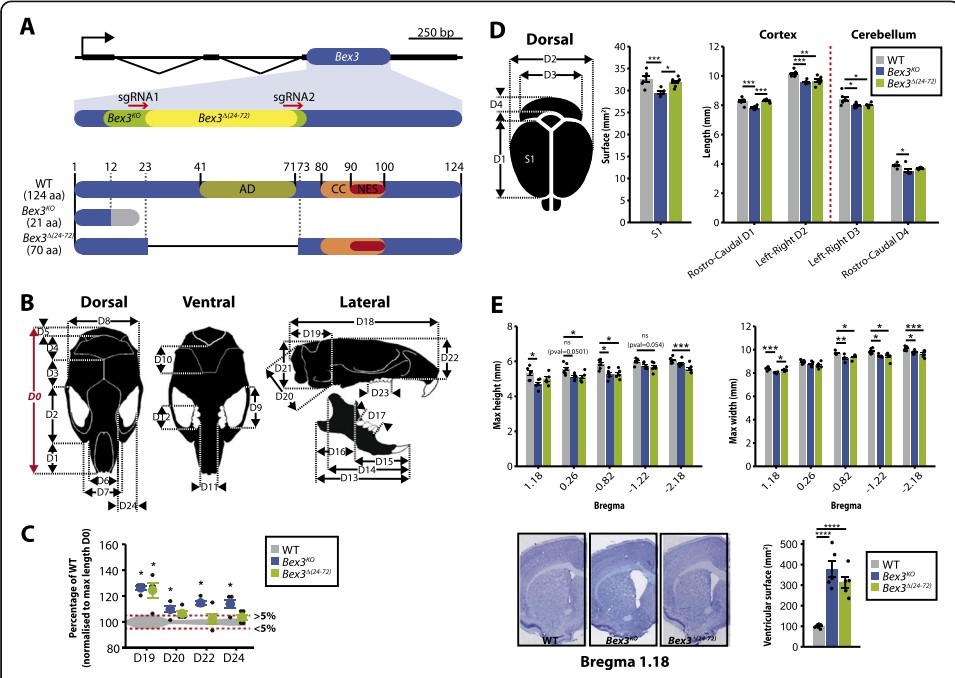

**Fig. 3** CRISPR-Cas9-generated *Bex3* mutant alleles show skull and brain abnormalities. **a** Schematic representation of the mouse *Bex3* locus (mm10; chrX:136,270,126-136,272,051) depicting exon-intron structure (non-coding and coding exons represented as black and blue rectangles, respectively). Two sgRNAs (red arrows) were used to generate CRISPR-Cas9-mediated deletions. The altered proteins potentially produced by the edited alleles that were selected to generate homozygous mice are shown below. In the *Bex3^KO* line, the deletion caused a frameshift in the open reading frame (region in gray) leading to the appearance of a premature STOP codon. The deletion in the *Bex3^Δ(24–72)* line removed 54 amino acids from the central core of the protein. AD, pro-apoptotic domain; CC, coiled coil domain; NES, nuclear export signal. **b**, **c** Morphometric analyses of skulls from wild-type and mutant *Bex3^KO* 6-week-old males employing a total of 24 anatomical measurements **b** revealed that *Bex3* dysfunction led to cranial abnormalities in frontal bone and skull height **c** (the complete analysis can be found in Additional file 1: Fig. S10). Measurements were normalized to maximum skull length (D0) and expressed relative to controls (black horizontal line). Deviations of 5% with respect to controls are shown as dotted red and green horizontal lines. Results are presented as mean ± SEM ($N \geq 4$); *$p < 0.05$, one-way ANOVA. **d**, **e** *Bex3* mutant brains showed altered brain morphology as evidenced by gross **d** and refined **e** anatomical measurements, concomitant with enlarged ventricular surfaces. Scale bar: 1 mm. Results in **d** and **e** are presented as mean ± SEM ($N \geq 5$); *$p < 0.05$, **$p < 0.01$, ***$p < 0.005$, ****$p < 0.001$, one-way ANOVA

structures of the cortex, hippocampus, and cerebellum, being preserved in the mutants (data not shown). In addition, we observed a marked increase in brain ventricular surface in both mutant lines (Fig. 3e). It is worth noting that reduced brain size and enlargement of brain ventricles have been described extensively in neurodevelopmental diseases [45–47].

### Behavioral defects in *Bex3* mutant mice

To determine if the anatomical and physiological brain defects observed in the *Bex3* mutant mice have an effect on animal behavior, adult mutant and control mice were subjected to a comprehensive battery of behavioral tests (Fig. 4 and Additional file 1: Fig. S11). In non-stressful open field, *Bex3* mutant mice showed normal locomotor activity [$F_{(10,1)} = 0.291$; $p = 0.601$ and $F_{(10,1)} = 1.628$; $p = 0.230$] for *Bex3^KO* and *Bex3^Δ(24–72)* lines respectively (Fig. 4a). However, we observed that both *Bex3^KO* and *Bex3^Δ(24–72)* lines displayed significantly more repetitive behavior events [rearing: *Bex3^KO*: $F_{(10,1)} = 5.078$; $p =$

0.047 and $Bex3^{\Delta(24-72)}$: F (10,1) = 1.355; $p$ = 0.271; grooming: $Bex3^{KO}$: F (10,1) = 6.188; $p$ = 0.032 and $Bex3^{\Delta(24-72)}$: F (10,1) = 22.216; $p$ < 0.001] when compared to their wild-type littermates (Fig. 4a). We evaluated social interaction using a three-chambered assay to study the interactions with familiar or stranger mice (Fig. 4b). While wild-type mice showed preferential interaction with familiar mice against the empty compartment [F (10, 1) = 94.218; $p$ < 0.001], we only found impairment in the interaction with the familiar mice in $Bex3^{KO}$ line, with no significant differences in the sniffing time between empty and the familiar mice containing compartments [F (10,1) = 2.053; $p$ = 0.182]. In contrast, in a social novelty paradigm, control animals spent significantly more time in close interaction with the novel animal, whereas $Bex3^{KO}$ displayed no significant preference for social novelty [F (10,1) = 3.114; $p$ = 0.108] or even avoided social novel interaction in $Bex3^{\Delta(24-72)}$ [F (10, 1) = 51.494; $p$ < 0.001]. Furthermore, only $Bex3^{KO}$ mice also showed impairment in nest building [F (10,1) = 9.800; $p$ = 0.010; Fig. 4c], which has been correlated with abnormal social organized behavior [48, 49]. Sensorimotor gating, whose impairment is also strongly associated with some neurodevelopmental disorders [50], was assayed by prepulse inhibition (PPI) of acoustic startle reflex. Although $Bex3$ mutants showed similar acoustic startle reflex than wild-type mice (Additional file 1: Fig. S11), both mutant lines exhibited a significant enhancement of PPI [$Bex3^{KO}$: F (10,1) = 6.588; $p$ = 0.028 and $Bex3^{\Delta(24-72)}$: F (10,1) = 6.038; $p$ = 0.033; Fig. 4d]. Moreover, cognitive behavior was assayed in the Y-maze test, object recognition memory, and passive avoidance tests. In the Y-maze test, $Bex3$ mutant mice showed significantly lower spontaneous alternation indexes, indicative of attention deficits and/or working memory [$Bex3^{KO}$: F (10,1) = 5.186; $p$ = 0.045 and $Bex3^{\Delta(24-72)}$: F (10,1) = 8.125; $p$ = 0.017; Fig. 4e]. On the contrary, only $Bex3^{KO}$ mice showed significant lower performance compared to their control littermates in object recognition memory [$Bex3^{KO}$: F (20,3) = 4.843; $p$ = 0.039 and $Bex3^{\Delta(24-72)}$: F (20,3) = 0.549; $p$ = 0.467, analyzed by 2-way ANOVA for genotype X session interaction; Fig. 4f] and passive avoidance tests [$Bex3^{KO}$: F (20,3) = 6.548; $p$ = 0.018 & $Bex3^{\Delta(24-72)}$: F (20,3) = 0.262; $p$ = 0.613, analyzed by 2-way ANOVA for genotype X session interaction; Fig. 4g]. In summary, the behavioral experiments indicated that both $Bex3$ mutant lines display repetitive behaviors and abnormal social conducts and that $Bex3^{KO}$ mice presented more severe phenotypes, especially in cognitive tests.

### Reduction in the number of cortical and subcortical interneurons in *Bex3*-deficient mice

A link between the disruption of interneuron inhibitory circuits in the neocortex and some clinical aspects of neurodevelopment and psychiatric disorders has already been established [51]. Our data show that both $Bex3$ mutant lines display gross brain morphological alterations and have abnormal behavioral traits and that $Bex3^{KO}$ mice presented more severe phenotypes, especially in terms of memory and learning impairment. These results, together with transcriptomic and structural genomic data related to human neurological disorders (Additional file 1: Tables S2 and S3) [40–42], suggest that $Bex3$ may function through the maintenance/renewal of specific neurons, and its absence can give rise to neurological conditions.

We measured parvalbumin-positive (PV+) interneuron density and found it was significantly decreased in motor, cingulate and sensory cortices, and caudate-putamen of $Bex3^{KO}$ mice; $Bex3^{\Delta(24-72)}$ mice showed a significant reduction of PV+ interneurons

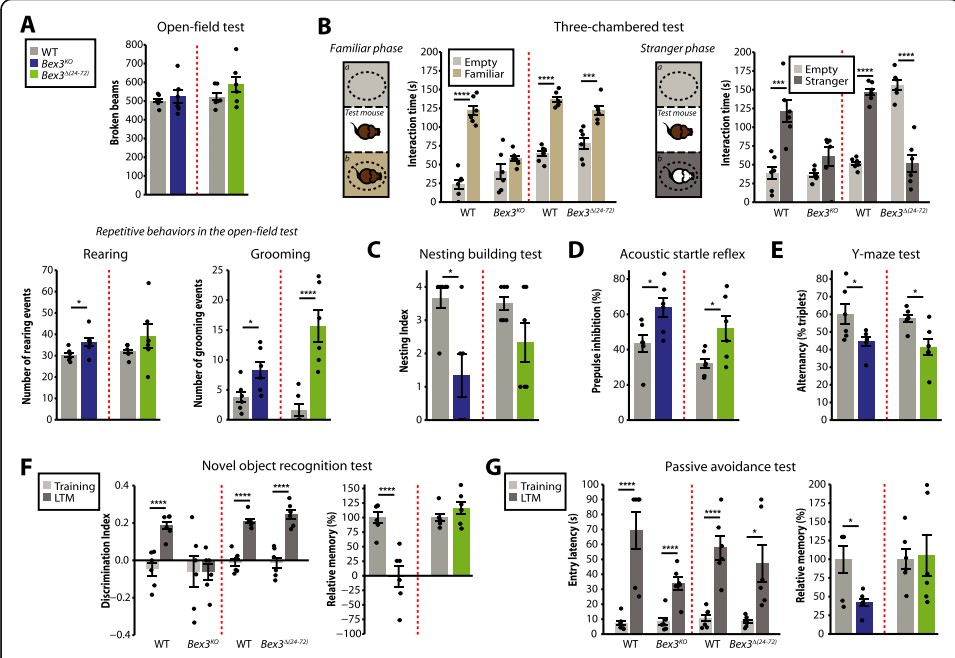

**Fig. 4** *Bex3* mutant mice display an array of behavioral and cognitive alterations. Adult control and *Bex3* mutant mice were subjected to a comprehensive battery of behavioral tests. **a** Locomotor activity (number of broken beams) and total repetitive behaviors (number of grooming and rearing) were evaluated in a 5-min open field test. **b** Social behavior was assayed in the three-chambered test. Sniffing times are showed in social familiar and social stranger phases. **c** Social organized behaviors were tested by the nesting building assay. **d** Sensory motor integration was determined by prepulse inhibition of acoustic reflex. **e** The Y-maze test was performed to evaluate repetitive conducts, attention, and/or working memory. As alternance, the % of triplets was determined. **f** Novel object recognition memory was evaluated 24 h after a training session. A positive discrimination index in this memory test indicates novel object preference exploration. **g** The passive avoidance test was used to evaluate emotional cognition. Entry latencies to the dark compartment in training and memory testing sessions are shown. Results are presented as mean ± SEM ($N = 6$ mice per experimental group); *$p < 0.05$, **$p < 0.01$, ***$p < 0.005$, ****$p < 0.001$, one-way ANOVA

only in cingulate and sensory cortices (Fig. 5a, b) and minor, non-significant, decrease in motor cortex and caudate-putamen. On the other hand, altered numbers of hippocampal neurons and interneurons have been associated to the onset of social and cognitive deficits in several animal models for neurological disorders (reviewed in [52]). Even though the hippocampus-to-hemisphere size ratio was not significantly altered in *Bex3* mutant mice (Fig. 5c), we observed a strong reduction in the density of total neurons and, particularly, calretinin-positive (CR+) inhibitory interneurons in the *stratum radiatum* of the CA1–2 hippocampal fields of *Bex3*$^{KO}$ mice (Fig. 5d, e).

We decided to analyze the status of the adult hippocampal neurogenic niche in *Bex3* mutant mice. Interestingly, the density of immature neurons was significantly decreased in the subgranular zone of *Bex3*$^{KO}$ mice, but not in *Bex3*$^{\Delta(24–72)}$ (Fig. 5f, g), thus suggesting that adult neurogenesis could be partially compromised in these animals.

## Alteration in the excitation/inhibition balance in the hippocampal CA2 region of *Bex3*$^{KO}$ mice

Impaired social interaction and social learning is known to rely on the CA2 neuronal circuit within the hippocampus [53, 54], while altered interneuron numbers in the CA2 hippocampal field can disrupt the excitatory/inhibitory hippocampal balance. Thus, we

tested if CA2 synaptic transmission was altered in *Bex3*$^{KO}$ mice, which have reduced numbers of hippocampal interneurons in this area. To determine whether the evoked synaptic activity was altered in hippocampal pyramidal CA2 neurons, we monitored evoked excitatory (eEPSCs) and inhibitory (eIPSCs) postsynaptic currents. We found that the amplitude of eIPSCs—but not eEPSCs—was clearly decreased in mutant mice (Fig. 5h). However, no statistically significant differences were found between wild-type and *Bex3*$^{KO}$ mice in the input-output curve, indicating that the net basal synaptic activity was not affected in these animals (Fig. 5i). Additionally, we studied spontaneous synaptic transmission onto hippocampal CA2 pyramidal neurons (Fig. 5j-m). Total spontaneous activity frequency (sEPSC + sIPSC) was significantly decreased in *Bex3*$^{KO}$ mice respect to wild-type (Fig. 5l) due to a strong reduction in sIPSCs frequency. On the other hand, no differences were found between sEPSCs and sIPSCs amplitude when comparing wild-type and mutant mice (Fig. 5m). These results indicate an excitation/inhibition imbalance in the hippocampal CA2 region of *Bex3*$^{KO}$ mice due to a strong decrease in inhibitory synaptic transmission, which is in agreement with the reduced density of inhibitory interneurons observed in this region.

### Altered mTOR signaling in the adult brain of *Bex3*-deficient mice

BEX3 protein physically interacts with the TSC1/TSC2 complex [55], which is essential for the proper regulation of the mTOR signaling cascade [56]. This pathway controls brain development and function at multiple levels, and its dysregulation has been implicated in several neurological diseases [57]. Intriguingly, we found a decrease in the phosphorylation of the mTORC1 readout S6K1 in brain lysates of both *Bex3* mutant lines, which would suggest mTORC1 hypoactivation; in contrast, no changes were detected in other targets like 4E-BP1 or S6. The mTORC2 readouts AKT and NDRG1 were hyperphosphorylated, whereas PKCα and PKCγ protein levels were unaltered (Fig. 6 and Additional file 1: Fig. S12). All these data suggest a hyperactivation of mTORC2 signaling in the adult brain as one of the possible molecular consequences linked to *Bex3* deficiency.

### Discussion

By screening the human and mouse genomes for domesticated transposons, we have identified the footprints of two L1 retrotransposons within the human *TCEAL7* gene, indicating that a domestication event restricted to placental mammals gave rise to a whole new gene family in the X chromosome. After the eutherian-specific gene L1TD1 [58], the *Bex/Tceal* family represents the second case of domestication of non-LTR transposons reported in metazoans, and the first to give rise to a multigenic family. This domestication generated an ancestral protogene that we termed *HALEX*, and our experiments in a non-eutherian model suggest an evolutionary scenario where this element was progressively integrated into anciently established gene networks through complete neofunctionalization before the diversification of eutherians. This is in contrast with most other known domestication cases, where new genes perform tasks at the cellular level similar to those of the ancestral element [59]. Several members of the *Bex/Tceal* gene family have been reported to code for hub proteins, due to their high number of interactions with multiple proteins belonging to several signaling pathways

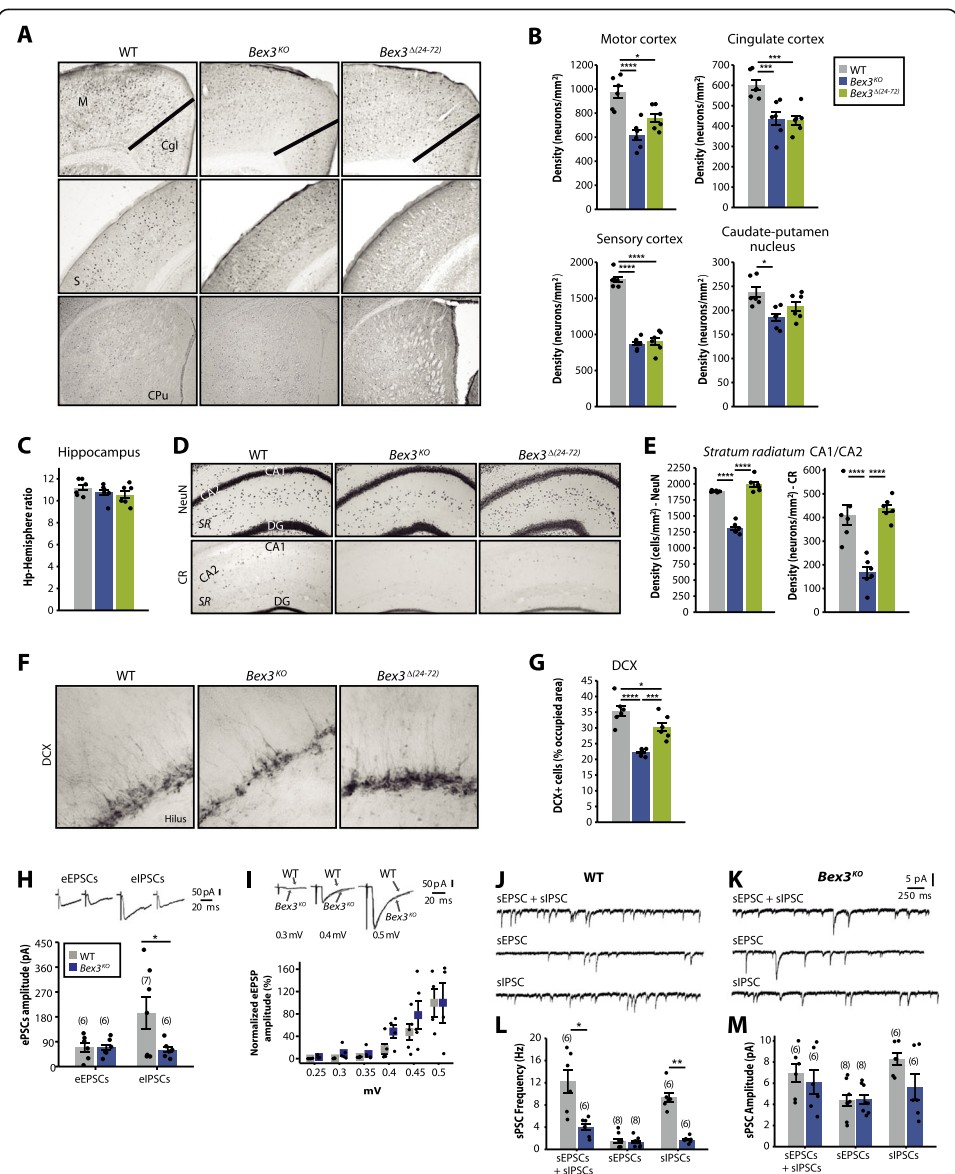

**Fig. 5** Adult hippocampal neurogenesis, cortical interneuron density, and excitation/inhibition balance are altered in *Bex3* mutant mice. **a** Representative microphotographies of PV immunoreactivity in motor (M), cingulate (Cgl), and sensory (S) cortices, and caudate-putamen (CPu) nucleus. **b** PV+ interneuron density quantification reveals significant differences among the groups. **c** Relative hippocampus area with respect to total area of coronal section was determined by NeuN immunostaining and showed no differences between animal groups. **d** Representative images of NeuN and CR immunostainings depicting hippocampal sections of control and *Bex3* mutant mice. *SR*; *stratum radiatum*. **e** A significant reduction in the number of NeuN+ neurons and calretinin (CR) interneurons in the *stratum radiatum* of hippocampal CA1–2 was found in *Bex3*^KO but not *Bex3*^Δ(24−72) mice. **f** Representative photomicrographies of *dentate gyrus* immunolabeled for DCX (doublecortin) as a marker for adult immature neurons. **g** % of DCX-positive area showing a significant and selective decrease in *Bex3*^KO mice. For morphological analyses, results are presented as mean ± SEM ($N = 6$ mice per experimental group); *$p < 0.05$, **$p < 0.01$, ***$p < 0.005$, ****$p < 0.001$, one-way ANOVA. **h** Evoked eEPSCs and eIPSCs in WT and *Bex3*^KO mice. **i** Relationship between the applied voltage and the amplitude magnitude of ePSCs from CA3-CA2 synapses in WT and *Bex3*^KO mice. **j**, **k** Representative recordings of spontaneous excitatory and inhibitory synaptic activity onto CA2 neurons illustrating frequencies and amplitudes of total (sPSCs), excitatory (sEPSCs), and inhibitory (sIPSCs) synaptic activity from WT **j** and *Bex3*^KO mice **k**. **l** Quantification of results in **j** and **k** in frequencies. **m** Quantification of results in **j** and **k** in amplitudes. Results are presented as mean ± SEM and the number of slices is shown in parentheses ($N = 6$–8 slices from 3 to 4 mice); *$p < 0.05$, **$p < 0.01$, Student's two-tailed *t* tests

[11, 12, 35, 60–64]. Remarkably, we found that both BEX/TCEAL proteins and the ancestral HAL1b sequence from which they derive are predicted to be predominantly disordered at their N-terminus and contain C-terminal α-helices (Additional file 1: Fig. S4). Relatedly, disorder is a structural property that frequently translates into variable conformations and the ability to bind multiple partners [65]. Thus, we suggest that the evolutionary potential of *Bex/Tceal* genes for a multiplicity of protein-protein interactions might be a direct legacy of the ancestral HAL1b retrotransposon protein.

The mammalian target of rapamycin (mTOR) pathway stands out as one of the molecular networks in which *Bex/Tceal* genes integrated, as shown by previous work on *BEX2* and *BEX4* [62, 64]. The mTOR protein kinase is involved in a myriad of processes related to cell growth, proliferation, and survival. It can assemble into two molecularly and functionally different complexes that include some specific components such as mTORC1 and mTORC2 [66]. mTORC1 main substrates are S6K and 4E-BP kinases, while mTORC2 downstream targets include AKT and SGK kinases. Upon phosphorylation, AKT inhibits the TSC1/2 complex [67], which regulates both mTORC signaling cascades in opposite directions: it inhibits mTORC1 and activates mTORC2. Yet, it can interact physically with mTORC2 only, specifically through Tsc2 [67].

Our data show that *Bex3* deficiency leads to hyperphosphorylation of Akt on Ser-473 in brain extracts, suggesting mTORC2 hyperactivation. Akt hyperphosphorylation should also induce hyperactivation of the mTORC1 route but no changes in 4E-BP1 phosphorylation were detected, while S6K1 was found to be hypophosphorylated on Thr-389. Importantly, earlier phosphorylation steps on the autoinhibitory domain of S6K1 are needed to render residue Thr-389 accessible to mTORC1. Both mTORC1 and JNK are involved in the phosphorylation of this domain of S6K1 in muscle [68]. JNK is inhibited by mTORC2, and therefore, mTORC2 hyperactivation in our models could indirectly lead to the observed S6K1 hypophosphorylation on Thr-389. Surprisingly, phosphorylation levels of S6K1 main effector S6 on residues Ser235/236 were not affected, suggesting the possible existence of compensatory mechanisms by other kinases that can target these residues [69]. In summary, our biochemical data and the fact that BEX3 interacts with the TSC1/TSC2 complex [55] suggest the putative involvement of the mTOR pathway in some of the reported phenotypic alterations. However, the weak differences found in some mTOR effectors imply that additional signaling pathways may also be involved in the reported changes.

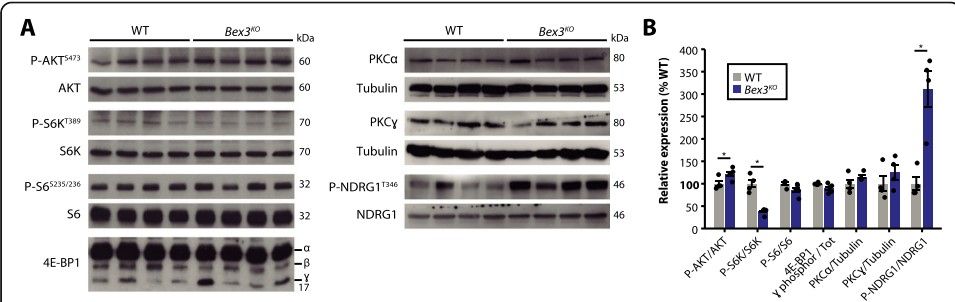

**Fig. 6** *Bex3* deficiency leads to aberrant mTOR signaling in the brain. **a** Western blot analyses of whole brain lysates of adult *Bex3*^KO^ mice revealed abnormal phosphorylation ratios of some mTORC1 and mTORC2 targets. Representative images. **b** Quantification of data in **a**, relative to wild-type. Results are presented as mean ± SEM (N = 4–5 per experimental group); *p < 0.05, non-parametric Mann–Whitney test

Given that BEX3 physically interacts with TSC1, an interaction shown to prevent proteasome degradation and thus necessary for NGF-p75NTR-BEX3-mediated apoptosis [55], we propose that BEX3 could prevent the TSC1/2 complex from interacting with mTORC2, eventually reducing the activity of the pathway. The absence of *Bex3* would then cause a hyperactivation of the mTORC2 pathway without affecting mTORC1. Thus, the function of BEX3 could be that of fine-tuning the regulation of these cascades. The fact that the $Bex3^{KO}$ and $Bex3^{\Delta(24-72)}$ lines show analogous results suggests that the missing central core of the mutant $Bex3^{\Delta(24-72)}$ protein is necessary for TSC1/2 recognition or, alternatively, that its coupling with the TSC1/2 complex does not interfere with mTORC2 binding, presumably due to the smaller size of the resultant protein. Furthermore, since mTOR signaling has been involved in adult neurogenesis and neuronal excitability and is dysregulated in several neurological diseases, the mTOR pathway is a compelling candidate to explain, at least in part, the molecular alterations behind the phenotype observed in *Bex3*-deficient mice.

The acquisition of regulatory sequences enabling transcription is also a crucial but scarcely studied step during the process of domestication and subsequent neofunctionalization of transposable elements [70]. We have described here a short, non-coding regulatory region called BGW motif that was co-opted during eutherian evolution by three unrelated gene ancestors generated from retrotransposition events.

Although transposons and mammalian-restricted genes predominantly show tissue-specific patterns [71, 72], we observed a relatively wide expression for most *Bex/Tceal* genes. In mouse embryos, expression profiles of *Bex*/*Tceal* genes are consistently associated with proliferative tissues within organs such as the stomach, lungs, pancreas, or central nervous system. Interestingly, some of these genes are used as markers for progenitor cells in developing tissues and are involved in cell proliferation, differentiation, and cell death [11, 29, 34, 73, 74]. In adult tissues, our results suggest that *Bex3* regulates adult hippocampal neurogenesis, while some *Bex/Tceal* genes are induced upon injury, having an impact on axonal, muscular, and hepatic regeneration [29, 37, 39].

*BEX1* and *BEX3* have been recently identified among the most significantly downregulated genes in excitatory neurons of the prefrontal cortex of Alzheimer's disease patients [75]. Furthermore, patients with severe intellectual disability, craniofacial dysmorphism, and autism have been reported to carry different genomic microdeletions in Xq22 encompassing *BEX/TCEAL* genes, with *BEX3* pinpointed as one of the main candidates to cause these neurological features [40–42]. Also, a small 252-kb duplication spanning *BEX3*, *TCEAL4*, *TCEAL9*, and *RAB40A* has been reported in a patient with autism (Decipher database, ID: 290829) [76]. Here we show that mutations of murine *Bex3* lead to subtle craniofacial changes and have a profound impact on repetitive and social behavioral performance, two important behavior alterations required to diagnose autism spectrum disorders (ASD) [77]. Consistent with ASD-like behaviors, some cerebral cortex areas, related with social behaviors [78, 79], and the striatum, related with repetitive behavior [80], showed alterations in the number of parvalbumin interneurons, another feature found in ASD mice models and patients [81–88]. Cognitive alterations are frequently found in ASD patients [89]. In this aspect, only $Bex3^{KO}$ mice show learning and memory defects in object recognition and passive avoidance tests, two hippocampus-dependent paradigms [90–92]. In parallel, alterations in calretinin interneurons in the CA1–2 hippocampal *stratum radiatum*, excitatory-inhibitory

imbalance in the CA2 field, and potential defects in adult *dentate gyrus* neurogenesis were also found only in $Bex3^{KO}$ mice. In this regard, the less severe phenotype of $Bex3^{\Delta(24-72)}$ mice suggests that the resulting protein may act as a hypomorph version of the wild-type allele. In brief, the phenotypical features of *Bex3* mutant mice makes this gene an exciting candidate for future research into human neurological disorders that impact upon repetitive behavior, sociability, and intellectual disability.

## Conclusions

We characterized the evolutionary process by which an ancient retrotransposon was inserted into the genome of an eutherian ancestor and progressively integrated into host molecular networks while acquiring crucial functions for the organism. We suggest that this process of strict neofunctionalization was channeled by the inherited structural properties of the ancestral transposable element, facilitating the evolution of protein interactions. This domesticated gene subsequently duplicated, generating the *Bex/Tceal* cluster, whose expression is enriched in neural tissues. By generating new alleles for one of its members, *Bex3*, we show that this gene is involved in the development, maintenance, and function of specific areas of the central nervous system. The mutant alleles show subtle anatomical alterations in skull morphology, and important variations in cortical and subcortical neuron populations, as well as electrophysiological hippocampal imbalance, which can explain some of the behavioral changes identified. We also find a disruption in the mTOR pathway that might explain the molecular cause underlying some of the observed phenotypes. It remains to be elucidated if other genes in the cluster, most of which are highly expressed in the central nervous system, play similar or complementary roles in the establishment of higher neurological functions in placental mammals, or into which existing signaling pathways have been incorporated.

## Methods

### Study design

Our original hypothesis posited that recently improved genome and repetitive element annotations may help uncover new events of transposon domestication in the human and mouse genomes. Thus, the primary objective was to identify previously undetected TE-derived genes in the human and mouse genomes. A bioinformatic genome-wide screening provided a list of putative candidates, including previously uncharacterized cases. We decided to study in depth the event that gave rise to the *Bex/Tceal* gene family, which originated at the base of the eutherian lineage. We hypothesized that they could have provided an evolutionary novelty and sought to functionally characterize one of its members, *Bex3*, to explore this possibility. Using the CRISPR/Cas9 technology in mice, we generated two mutant lines for this gene, namely $Bex3^{KO}$ and $Bex3^{\Delta(24-72)}$, which were maintained in a CBA/C57Bl6 hybrid background. These mice, together with wild-type counterparts, were subjected to a battery of morphological, behavioral, physiological, histological, and molecular analyses. Additionally, chick embryos were employed as a non-eutherian animal model. All experiments were performed using 4 to 7 animals per experimental group, as presented in the manuscript. They were randomly chosen from the available animals in each case. For quantitative data, most

measurements were made automatically and, thus, not subject to operator bias. Additionally, the limited number of researchers working on each experiment prevented blinding procedures. No data were excluded from the analysis.

### Identification of domesticated transposon candidates

Gene and repetitive element annotations for the hg38 human and mm10 mouse genome assemblies were downloaded from the table browser tool of the UCSC website, selecting UCSC genes and RepeatMasker tracks, respectively. In order to discard non-TE elements, we filtered the RepeatMasker output as in [93]. These gene and repetitive element annotations were directly compared, and the results were filtered according to the following criteria: (i) overlap above 50% across the candidate gene coding region; (ii) txCdsPredict score above 800, which is approximately 90% predictive of protein-coding genes (it considers the length of the ORF, the presence of a Kozak consensus sequence, nonsense mediated decay mechanisms, and upstream ORFs); and (iii) conservation of the ORF in more than one species. Mammalian species with genome available at Ensembl genome browser (http://www.ensembl.org) were considered when checking for ORF conservation.

### Sequence retrieval

BLASTn and BLAT searches in NCBI and UCSC databases using human sequences were carried out in order to identify *Bex/Tceal* family genes in eutherian species. To clarify the orthology between genes undergoing gene conversion, surrounding intergenic sequences were added to the corresponding human query. The genomic assemblies used were as follows: GRCh38/hg38 for human (*Homo sapiens*); Broad CanFam3.1/canFam3 for dog (*Canis familiaris*); GRCm38/mm10 for mouse (*Mus musculus*); Bos_taurus_UMD_3.1.1/bosTau8 for cow (*Bos taurus*); Broad/equCab2 for horse (*Equus caballus*); Baylor/dasNov3 for nine-banded armadillo (*Dasypus novemcinctus*); Broad/choHof1 for Hoffmann's two-toed sloth (*Choloepus hoffmanni*); Broad v1.0/triMan1 for Florida manatee (*Trichechus manatus latirostris*); Broad/monDom5 for gray short-tailed opossum (*Monodelphis domestica*); WTSI Devil_ref v7.0/sarHar1 for Tasmanian devil (*Sarcophilus harrisii*), TWGS Meug_1.1/macEug2 for tammar wallaby (*Macropus eugenii*). The consensus sequences of TE subfamilies were retrieved from the RepBase-derived RepeatMasker library update 20170127 (http://www.girinst.org/server/RepBase/).

### Evolutionary analyses and secondary structure prediction

Nucleotide and protein sequences were aligned using the L-INS-i iterative refinement method of MAFFT [94], and the resulting alignments were edited with Jalview [95]. The phylogenetic reconstruction was performed using IQ-TREE [96] and built-in software ModelFinder [97]. Branch support was calculated running 1000 replicates of the SH-like approximate likelihood ratio test [98] and ultrafast bootstrap [99]. Phylogenetic trees were visualized and edited with FigTree v1.4.2 [100] and Dendroscope 3 [101]. PCOILS was used for coiled coil prediction [102]. PSI-PRED 4.0 [103, 104] and DIS-OPRED3 [105] were used for α-helix and protein disorder prediction, respectively. Finally, the positive selection analysis was performed using HyPhy [106], specifically the

MEME [107] and aBSREL [108] methods. Taking into account the divergence observed between the *Bex* and *Tceal* subfamilies, the analysis was split in two parts to avoid saturation of substitutions.

### Differential expression analysis of eutherian adult tissues

RNA-seq data from eight homologous adult organs of dog, cow, and nine-banded armadillo were downloaded from SRA database: SRP016501 (GSE41637, *Bos taurus*), SRP114662 (GSE20113, *Canis familiaris*), and SRP012922 (GSE106077, *Dasypus novemcinctus*). Protein-coding cDNA sequences were downloaded from Ensembl (http://www.ensembl.org) as a transcriptomic index to map against. When *Bex/Tceal* orthologs were found to be incomplete or absent in these files, we replaced the partial sequences or introduced the missing ones, respectively. RNA-seq data was trimmed to 50 base pairs and mapped using Bowtie (allowing no more than two mismatches and discarding reads that mapped more than once) against their respective libraries for each species, and expression levels were calculated correcting for the effective length of each transcript (read-long positions repeated in other transcripts were excluded) to obtain cRPKM metrics [109]. cRPKM values for human and mouse were obtained from VAST DB [110]. Heatmaps were produced with Heatmapper [111] using Pearson's distance measurement and average linkage as the clustering method.

### Expression of *BEX/TCEAL* genes in autism spectrum disorder and schizophrenia

Differential expression of *BEX/TCEAL* genes was assessed using transcriptomic data from all publicly available human transcriptomic datasets for schizophrenia and autism spectrum disorder, either in GEO (http://www.ncbi.nlm.nih.gov/geo) or published articles (Additional file 1: Table S2). Statistical analysis of these data is specified in the corresponding "Statistical analysis" section.

### In situ hybridization

Mouse embryos were collected at E13.5, cryoprotected, and cut in 20-μm-thick sections. Primer pairs (Additional file 1: Table S4) were designed in order to amplify murine *Bex/Tceal* genes by PCR using cDNA. Next, digoxigenin-labeled RNA antisense probes were synthesized, and in situ hybridization was performed as described elsewhere [112]. Images were obtained with a Leica MZ16 F stereomicroscope.

### Generation of transgenic mice using the CRISPR-Cas9 system

Two CRISPR guide RNAs (sgRNAs) targeting the mouse *Bex3* gene were designed on exon 1 using CRISPR DESIGN (http://crispr.mit.edu) and CRISPRSCAN [113], considering potential off-target effects and predicted functional activity. They were generated by in vitro transcription from sgRNA DNA templates as previously described [114]. Briefly, for each sgRNA, two complementary oligos containing the CRISPR-Cas9 target sequence were annealed leaving BbsI-compatible overhangs. Oligo sequences for each sgRNA can be found in Additional file 1: Table S4. Annealed products were then cloned into the pgRNAbasic plasmid, a kind gift from Dr. Moises Mallo (Instituto Gulbenkian de Ciencia, Portugal), at the *Bbs*I site located downstream the T7 promoter and upstream the universal tracrRNA sequence necessary for sgRNA folding and

activity. Plasmids were linearized with *Fsp*I and the purified product used for T7 in vitro transcription according to the manufacturer's instructions (New England Biolabs). sgRNAs were purified by phenol:chloroform extraction after DNaseI treatment (Roche). Next, in vitro digestion assays were performed to evaluate sgRNA activity [115]. In brief, a DNA template containing the CRISPR-Cas9 target sites was amplified by nested PCR with the primers listed in Additional file 1: Table S4. Purified PCR products (200 ng) were incubated for 3 h at 37 °C with Cas9 protein (20 ng/μl, Addgene vector #47327) and sgRNA (2.5 ng/μl) in digestion buffer (20 mM HEPES pH 7.5, 150 mM KCl, 0.5 mM DTT, 0.1 mM EDTA, 10 mM $MgCl_2$). Cleavage efficiency was then analyzed by electrophoresis in a 1% agarose gel stained with ethidium bromide.

Cas9 mRNA (100 ng/μl, SBI) and sgRNA (10 ng/μl) were co-injected into the cytoplasm of CBA/C57Bl6 fertilized eggs using standard methods. Deletions in $F_0$ pups were detected by nested PCR (see Additional file 1: Table S4) of genomic DNA obtained from tail biopsies, and confirmed by sequencing after TA cloning into pCR2.1 (Invitrogen). Mutant $F_0$ carriers were crossed with wild-type CBA/C57Bl6 hybrids, and their descendants were likewise analyzed to identify the specific deletion allele transmitted. Mutant lines with deletions in mm10 chrX:136271359-136271505, for $Bex3^{\Delta(24-72)}$, and in chrX:136271327-136271522, for $Bex3^{KO}$, were established and maintained in a hybrid background.

### RT-PCR analyses

Brain RNA was extracted from three adult male wild-type and mutant mice using Tri-Pure isolation reagent (Roche), and cDNA was then synthesized with the first-strand cDNA synthesis kit for RT-PCR (AMV) (Roche), according to the manufacturer's instructions. *Bex3* and *actin* expression were analyzed with the primers listed in Additional file 1: Table S4.

### Morphometric analysis of mouse skulls

Control and mutant adult male mice at 6 to 8 weeks of age were sacrificed by cervical dislocation. Their heads were severed, skinned, defleshed, and incubated for 2–3 days in 2% NaOH/PBS in agitation to remove all remaining soft tissue. Undigested tissues where removed with forceps and skulls washed 3 times in PBS for 20 min. Samples were oriented and photographed using a high-resolution 3-CCD JVC camera (model KY-F55B) fitted onto a Nikon SMZ1500 stereomicroscope. Images were assembled, and 25 anatomical measurements were taken in Adobe Photoshop. Results were normalized to maximum skull length (D0 in Fig. 3) and relativized with respect to controls.

### Histological staining

Adult control and *Bex3* mutant mice were anesthetized and transcardially perfused for 20 min with 4% PFA/PBS. Brains were removed, post-fixed 24–48 h in 4% PFA/PBS, cryoprotected with 30% sucrose/PBS overnight, frozen in isopentane, and stored at −80 °C until use. Brains were sectioned coronally (30 μm), sections were collected in cryoprotectant solution (85% glycerol, 100% ethylene glycol, 0.1 M PBS) and kept at −20 °C until analysis. In the case of Nissl staining, sections were dehydrated and

mounted (Eukitt). Slide images were captured using a Nanozoomer slide scanner (Hamamatsu) and analyzed by measuring the maximum width (mm) and height (mm) in 5 sections per animal, corresponding to different anterior-posterior levels based on Bregma. Measurements were performed by using NanoZoomer Digital Pathology software. Ventricular surface was measured in 2 histological sections per animal by using ImageJ software. Additionally, immunohistochemistry (IHC) labelling was performed on 50-µm coronal brain cryosections with the following primary antibodies: mouse anti-NeuN (1:3000, Millipore), rabbit anti-parvalbumin (1:3000, Swant), mouse anti-calretinin (1:1000, Novocastra), rat anti-GFAP (1:3000, Calbiochem), and goat anti-doublecortin (1:500, Santa Cruz). Immunoreactivity was developed with DAB-peroxidase reaction. To minimize variability, at least 2–3 sections from each area were analyzed per animal on a bright-field DMRB RFY HC microscope (Leica). In each section, the total number of labeled cells per area of tissue was quantified using ImageJ software.

Chick embryos were fixed for 2–4 h at 4 °C in 4% (w/v) paraformaldehyde in PB, washed in PBS, and vibratome sectioned (45 µm). BrdU detection and immunostaining were performed following standard procedures [116, 117]. The following antibodies were used: rabbit anti-GFP (1:500, Invitrogen), rat anti-BrdU (1:500, AbDSerotec), rabbit anti-Sox2 (1:500, Invitrogen), Alexa488- and Alexa562-conjugated antibodies (Invitrogen). The sections were recorded using a Leica SPE confocal microscope. Cell counting was carried out on 10–17 pictures obtained from 5 to 7 chick embryos per experimental condition.

### Behavioral tests

Behavioral tests were performed in 3- to 5-month-old mice in a room with constant sound and light after 1 h of habituation. In all tests, during mice manipulation and behavioral evaluation, the researcher was blind to mice genotypes. The order in which the tests were run was always the same: open field, three-chambered test, nesting test, startle, Y-maze, object recognition memory, and step-down passive avoidance.

#### Open field test

Motor activity was assessed in an open field for 5 min (38 × 21 × 15 cm: Cybertec S.A., Madrid, Spain) as previously described [92]. This apparatus, which is coupled with infrared (IR) emitters and sensors connected to a computer, allowed to quantify the number of times mice interrupted the IR beams/min. In parallel, sessions were video recorded in order to evaluate rearing and grooming.

#### Three-chambered sociability test

Sociability was analyzed in an arena (60 × 60 cm) divided in three equal compartments. Each lateral compartment contained a clear Plexiglas cylinder (each 7 cm in diameter, 12 cm tall) with multiple holes (1 cm in diameter) to allow auditory, visual, and olfactory interaction between the stimulus mouse and the test mouse placed inside and outside of the cylinder, respectively. The paradigm, lasting 5 min per animal, consisted in a three-stage procedure: During the *habituation phase*, the test mouse was allowed to explore the

apparatus with the cylinders empty. In the *social familiar subject phase*, a conspecific from its same home cage was placed in one of the cylinders while the other remained empty. Finally, the *social stranger subject phase* was performed with an unfamiliar age- and sex-matched CD1 mouse placed in one of the cylinders maintaining the other empty. Social interaction was evaluated by the time that test mice spent sniffing each cylinder.

### Nesting test
The ability to build a nest was assessed following previously described procedures [118].

### Acoustic startle reflex and prepulse inhibition
During training, the mouse was placed in the startle chamber (Cibertec S.A., Madrid, Spain) for 3 min for acclimation; then, baseline startle responses were measured and averaged from 25 to 30 recordings after the presentation of 20 sounds (125 dB, 100 ms long). From this phase, the average response and peak latencies, as well as the peak response, were determined. During prepulse inhibition (PPI) trials, the same sound was preceded (250 ms) by a prepulse stimulus of 85 dB, 50 ms long. Trials including prepulse stimuli were randomly presented with normal startle stimuli, the final total being 25 of each, and the proportion of PPI was determined as [(1 – prepulse/startle) × 100]. The ambient background noise was 70 dB.

### Y-maze
Each mouse was placed in the center of a Y-maze consisting of three equally sized arms (8 × 40 × 20 cm), with white opaque walls at a 120° angle from each other, and allowed to freely explore the arms during 5 min. The number of arm entries and triads were used to calculate the alternation index. An entry was considered to occur when all four limbs were within the arm.

### Object recognition memory
The object recognition protocol was described extensively elsewhere [92]. Briefly, two equal objects were placed in a rectangular arena (55 × 40 × 40 cm) during the training phase. The next day, one object was replaced by a novel one, and the animal's memory of the original object was assessed by comparing the amount of time spent actively exploring the novel object against that for the familiar one using a discrimination index [DI = $(t_{novel} - t_{familiar})/(t_{novel} + t_{familiar})$). Exploration of an object was defined as directing the nose toward the object at a distance of ≤ 1.5 cm or touching the object with the nose or vibrissae. Circling or sitting on the object were not considered exploratory behaviors.

### Step-through passive avoidance test
During habituation, mice were allowed to explore for 1 min a chamber (47 × 18 × 26 cm, Ugo Basile) symmetrically divided into one light and one dark compartment separated by a door. During the training phase, mice were confined to the light compartment for 30 s and then allowed to access the dark compartment. Once inside, the door was closed automatically and the mice received an electrical stimulation (0.3 mA, 5 s) through the metal floor. Retention tests were performed the next day. Here, the latency

to enter into the dark compartment (escape latency) was assessed as a measure of memory retention.

### Electrophysiological recordings

Hippocampal slices were prepared as previously described [119, 120]. Five-month-old mice were anesthetized with isoflurane (2%) and decapitated for slice preparation. Briefly, after decapitation, the whole brain containing the two hippocampi, was placed into ice-cold solution (I) consisting of the following (in mM): 126 NaCl, 3 KCl, 1.25 $KH_2PO_4$, 2 $MgSO_4$, 2 $CaCl_2$, 26 $NaHCO_3$, and 10 glucose (pH 7.2, 300 mOsm), positioned on the stage of a vibratome slicer and cut to obtain transverse hippocampal slices (350 μm). Slices were maintained continuously oxygenated for at least 1 h before use. All experiments were carried out at room temperature (22–25 °C). For experiments, slices were continuously perfused with the solution described above. Whole-cell patch clamp recording of pyramidal cells located in the CA2 field of the hippocampus was obtained under visual guidance by IR differential interference contrast (DIC) microscopy and were verified as pyramidal cells through their characteristic voltage response to a current step protocol. Neurons were recorded in the current-clamp configuration with a Multiclamp 700B patch clamp amplifier. Data were acquired using pCLAMP 10.2 software (Molecular Devices). To record evoked excitatory postsynaptic currents (eEPSCs), electrical pulses were delivered to Schaffer collateral axons. To evoke inhibitory postsynaptic currents (eIPSCs), electrical pulses were delivered to interneurons situated in the *stratum oriens*. Spontaneous synaptic activity (sPSCs), either excitatory (sEPSCs) or inhibitory (sIPSCs), was recorded from hippocampal CA2 neurons.

Patch electrodes were pulled from borosilicate glass and had a resistance of 4–7 MΩ when filled with the following (in mM): 120 CsCl, 8 NaCl, 1 $MgCl_2$, 0.2 $CaCl_2$, 10 HEPES, 2 EGTA, and 20 QX-314 (pH 7.2, 290 mOsm). Experiments were performed at − 70 mV. Only cells with a stable resting membrane potential of − 55 mV were assessed, and the cell recordings were discarded if the series resistance changed by more than 15%. All recordings were low-pass filtered at 2 kHz and acquired at 10 kHz. Excitatory postsynaptic currents (evoked, eEPSCs, and spontaneous, sEPSCs) were isolated by adding bicuculline (20 μM) to the perfusion solution to block GABAA receptors. Inhibitory postsynaptic currents (evoked, eIPSCs, and spontaneous, sIPSCs) were isolated adding D-AP5 (50 μM) and NBQX (10 μM) to the perfusion solution to block NMDA receptor and AMPA/Kainate receptor-mediated currents, respectively. Signals were averaged every 12 traces. Spontaneous recordings consisted of 60 sweeps/5 s long that were analyzed for amplitude and frequency of detected events.

### Western blotting

Total brain tissue was homogenized in RIPA buffer (50 mM Tris-HCl pH 7.4, 150 mM NaCl, 1 mM EDTA, 1% (v/v) Triton X-100, 0.1% (w/v) SDS), containing protease inhibitor cocktail (Roche) and phosphatase inhibitors (2 mM sodium orthovanadate, 1 mM sodium pyrophosphate, 10 mM sodium fluoride). Protein concentration was measured using the bicinchoninic acid (BCA) protein assay as specified by the manufacturer (Pierce, Thermo Fisher Scientific). Samples were resolved by SDS-polyacrylamide

gels and transferred onto nitrocellulose membranes. Blocking and incubation with primary and secondary antibodies were performed following the manufacturer instructions. Membranes were developed with the ECL system (GE Healthcare), and acquired images were quantified using ImageJ software. Tubulin or GAPDH loading controls were used when needed. The primary antibodies used were the following: S6 1/1000 (#2217; Cell Signaling), p-S6 (Ser235/236) 1/1000 (#2217; Cell Signaling), S6K1 1/1000 (#9202; Cell Signaling), p-S6K1 (Thr389) 1/1000 (#9205; Cell Signaling), 4E-BP1 1/1000 (#9644; Cell Signaling), AKT 1/1000 (C-20, Santa Cruz Biotechnology), p-AKT (Ser473) 1/1000 (#9271; Cell Signaling), PKCα 1/1000 (10/2018, Cell Signaling), PKCγ 1/1000 (C-4, SC-166385, Santa Cruz Biotechnology), NDRG1 1/1000 (B-5, SC-398291, Santa Cruz Biotechnology), p-NDRG1 (Thr346) 1/1000 (#3217; Cell Signaling), GAPDH (6C5, ab 8245, Abcam), and tubulin 1/1000 (#9026; Sigma-Aldrich). The secondary antibodies used were HRP-labeled anti-mouse 1/2000 (P447-01, Vector) and anti-rabbit 1/2000 (P217-02, Vector).

### HALEX ancestral protogene reconstruction and in ovo electroporation

In order to obtain an approximate reconstruction of the ORF of the ancestral *HALEX* protogene, a segment of the consensus sequences of HAL1b and inverted L1MEe derived from RepeatMasker, together with the sequence of the *TCEAL7* human gene corresponding to the hypothetical target site duplication of an ancestral L1ME element, were combined (Additional file 1: Table S4). Finally, the synthesis of an artificial gene cloned into a pcDNA™3.1/*myc*-His was ordered to GenScript.

Eggs from white-Leghorn chickens were incubated at 38.5 °C in an atmosphere with 70% humidity and staged according to the method of Hamburger and Hamilton [121]. In ovo electroporation was performed at stage HH11-12 (48 h of incubation) with DNA plasmids as described previously [117, 122]. Bromo-deoxyuridine (1 mM; Sigma) was injected into the lumen of the chick neural tube at 20 min before harvesting, to label dividing cells. The embryos were recovered at 24 h post-electroporation.

### Statistical analysis

Enrichment analyses for *BEX* and *TCEAL* genes in gene expression datasets for autism spectrum disorder and schizophrenia were performed for each dataset using a hypergeometric test; $p < 0.05$ was considered as a threshold for statistical significance. Other statistical analyses were performed using Student's two-tailed $t$ tests or one-way ANOVA with Tukey's multiple comparison tests, where appropriate. When data were not normally distributed, non-parametric Mann–Whitney tests were used to determine the statistical significance. Calculations were performed with GraphPad Prism version 6 or in R; *$p < 0.05$, **$p < 0.01$, ***$p < 0.005$, ****$p < 0.001$.

## Supplementary information

**Additional file 1:** Fig. S1 The *Tceal7* gene is derived from the domestication of L1 retrotransposon fragments. Fig. S2 The *Bex/Tceal* gene cluster was established before the diversification of extant eutherians. Fig. S3 Highly diverged BEX/TCEAL proteins share a coiled coil domain. Fig. S4 BEX/TCEAL proteins might have inherited some of their structural properties from the ancestral transposon. Fig. S5 Selection pressure analyses reveal signatures of positive selection in the *Bex/Tceal* genes. Fig. S6 A BGW-like sequence was already present in the *GLA* promoter of the last therian common ancestor. Fig. S7 *Bex/Tceal* genes show tissue-enriched expression patterns during

development. Fig. S8 *Bex3* and *Tceal7* genes, but not the ancestral HALEX element, induce cell proliferation in chicken neural tube. Fig. S9 The deletions introduced using CRISPR-Cas9 technology can be observed in the mRNA expressed from the *Bex3* mutant alleles. Fig. S10 CRISPR-Cas9-generated *Bex3* mutant alleles show subtle skull abnormalities. Fig. S11 *Bex3* mutant mice show normal acoustic startle reflex. Fig. S12 *Bex3* deficiency leads to aberrant mTOR signaling in the brain. Table S1 Coding genes putatively derived from transposable elements in the human and mouse genomes. Table S2 Altered expression of *BEX* and *TCEAL* genes in subjects with autism spectrum disorder or schizophrenia. Table S3 Enrichment of differential gene expression in *BEX* and *TCEAL* gene families in subjects with autism spectrum disorder or schizophrenia. Table S4 Primers and reconstructed gene sequences used in this work. Supplementary references.

**Additional file 2.** Review history.

## Acknowledgments
We wish to thank Claudia Pérez and Alba del Valle for technical support, Jose Ramón Bayascas for help with western blots, Alejandro Sanchez-Gracia and Julio Rozas for advice on selection analyses, and Manuel Irimia and Ignacio Maeso for helpful discussions.

## Peer review information
Kevin Pang was the primary editor on this article and managed its peer review and editorial process in collaboration with the rest of the editorial team.

## Review history
The review history is available as Additional file 2.

## Authors' contributions
E.N.P. designed the study, performed molecular biology and in situ hybridization experiments, did bioinformatics and histologic analyses, and wrote the manuscript. C.V.G. designed the study, performed molecular biology experiments, and wrote the manuscript. S.M. did molecular biology experiments, performed histological analyses, and wrote the manuscript. D.B. designed the study, performed bioinformatics analyses and molecular biology experiments, and wrote the manuscript. N.F.C. did bioinformatics analyses. J.L.F. supervised in situ hybridization experiments in mice. M.L.M. did all skull measurements and preliminary behavioral tests. M.A.N. performed behavioral tests. I.S.-P. performed behavioral tests. E.A.G. did bioinformatics analyses. F.U. supervised molecular biology analyses. C.H.U. conducted selection pressure analysis. P.C. performed bioinformatics analyses. R.F.M. performed excitation/inhibition balance experiments. A.R.M. supervised excitation/inhibition balance experiments. S.D'A. helped in writing the manuscript. B.C. supervised bioinformatics analyses and provided resources. G.M. supervised molecular biology experiments and provided resources. E.S. supervised molecular biology and histology experiments and provided resources. A.M.C. performed and supervised behavioral and histology experiments and provided resources. J.J.C. designed the study, generated the mouse mutant lines, performed and supervised molecular biology experiments, and provided resources. J.G.F. designed the study and provided resources and supervision. The author(s) read and approved the final manuscript.

## Authors' information
Twitter handles: @cvicgar (Cristina Vicente-García); @MirraSerena (Serena Mirra); @Noeliafdez6 (Noèlia Fernàndez-Castillo); @jlferran (José Luis Ferrán); @maclopmay (Macarena López-Mayorga); @yomellamocar (Carlos Herrera-Úbeda); @Salv_DANIELLO (Salvatore D'Aniello); @bcormand (Bru Cormand); @GMarfanyN (Gemma Marfany); @jcarvajal (Jaime J. Carvajal); @jordigarciafdez (Jordi Garcia-Fernàndez).

## Funding
Major financial support for this research was received from Spanish "Ministerio de Ciencia, Innovación y Universidades." Grants BFU2015-68655-P and BFU2017-861152-P to J.G.F., RTI2018-100968-B-I00, 2017-SGR-738, H2020/2014-2020 under grant agreements n°667302, n°643051, and n°728018 to B.C., PGC2018-098229-B-I00 to J.L.F., BES-2016-077374 to E.A.-G., CVI-7290 Junta de Andalucía to A.R.M., SAF2016-80937-R (Ministerio de Economía y Competitividad/FEDER) to G.M., Institutional Grant MDM-2016-0687 (Maria de Maeztu Excellence Unit, Department of Gene Regulation and Morphogenesis at CABD) and BFU2017-83150-P to J.J.C, BFU2017-89780-R and P12-CTS-2257 to A.M.C. and SAF2016-76340-R and María de Maeztu Excellence Unit, Institute of Neurosciences to E.S.. E.N.P. held an FPI pre-doctoral fellowship (Spanish "Ministerio de Ciencia, Innovación y Universidades"). S.M. was first supported by a contract with the "Centro de Investigación Biomédica en Enfermedades Neurodegenerativas," and later by "Centro de Investigación Biomédica en Red de Enfermedades Raras" (CIBERER). N.F.C. is also under contract by CIBERER. This study makes use of data generated by the DECIPHER community. A full list of centres who contributed to the generation of the data is available at http://decipher.sanger.ac.uk and via email from decipher@sanger.ac.uk. Funding for the project was provided by the Wellcome Trust.

## Availability of data and materials
All data generated during this study are included in this published article and its supplementary information files. All data reanalyzed in this study are publicly available from the sources specified in the methods. Specifically, RNA-seq datasets SRP016501 (GSE41637, *Bos taurus*), SRP114662 (GSE20113, *Canis familiaris*), and SRP012922 (GSE106077, *Dasypus novemcinctus*) were downloaded from SRA database, while protein-coding cDNA sequences were obtained from Ensembl (http://www.ensembl.org), and cRPKM values for human and mouse tissues, from VASTDB [110]. Further, human transcriptomic datasets for schizophrenia and autism spectrum disorder were obtained from GEO database (GSE38322, GSE35978, GSE53987, GSE87610) or published articles (Additional file 1: Supplementary references [16–18, 21–26]).

## Ethics approval and consent to participate

Experiments using animals were performed under protocols approved by the Universidad Pablo de Olavide Ethical Committee (Sevilla, Spain; protocols 24/04/2018/056 and 24/04/2018/043) and the Ethical Committee for Animal Experimentation (AEC) of the Generalitat of Catalonia (protocol 9431) in accordance with Spanish Royal Decree 53/ 2013, European Directive 2010/63/EU, and other relevant guidelines.

## Consent for publication

Not applicable.

## Competing interests

The authors declare no competing interests.

## Author details

[1]Department of Genetics, Microbiology and Statistics, Faculty of Biology, and Institut de Biomedicina (IBUB), University of Barcelona, 08028 Barcelona, Spain. [2]Centro Andaluz de Biología del Desarrollo, CSIC-UPO-JA, Universidad Pablo de Olavide, 41013 Sevilla, Spain. [3]Department of Cell Biology, Physiology and Immunology, and Institute of Neurosciences, University of Barcelona, 08028 Barcelona, Spain. [4]Centro de Investigación Biomédica en Red de Enfermedades Raras (CIBERER), Instituto de Salud Carlos III (ISCIII), Madrid, Spain. [5]Centro de Investigación Biomédica en Red sobre Enfermedades Neurodegenerativas (CIBERNED), Instituto de Salud Carlos III (ISCIII), 28029 Madrid, Spain. [6]Department of Zoology, Charles University, Vinicna 7, 12844 Prague, Czech Republic. [7]Institut de Recerca Sant Joan de Déu (IR-SJD), Esplugues de Llobregat, 08950 Barcelona, Spain. [8]Department of Human Anatomy, School of Medicine, University of Murcia and IMIB-Arrixaca Institute, 30120 Murcia, Spain. [9]Department of Physiology, Anatomy and Cell Biology, Universidad Pablo de Olavide, 41013 Sevilla, Spain. [10]Present Address: Instituto de Neurociencias de Alicante (Universidad Miguel Hernández - Consejo Superior de Investigaciones Científicas), Alicante, Spain. [11]Present Address: Centro de Investigación Biomédica en Red de Salud Mental (CIBERSAM), Neuropsychopharmacology and psychobiology research group, UCA, INiBICA, Cádiz, Spain. [12]Genome Architecture, Gene Regulation, Stem Cells and Cancer Programme, Centre for Genomic Regulation (CRG), the Barcelona Institute of Science and Technology, 08003 Barcelona, Spain. [13]Universitat Pompeu Fabra (UPF), 08003 Barcelona, Spain. [14]Department of Biology and Evolution of Marine Organisms, Stazione Zoologica Anton Dohrn, 80121 Naples, Italy. [15]Institució Catalana de Recerca i Estudis Avançats (ICREA), 08010 Barcelona, Spain.

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

## 