## [**Additional file 2.** Review history. · Genome Biology]

Review History

First round of review

Reviewer 1

Are you able to assess all statistics in the manuscript, including the appropriateness of statistical tests used? Yes, and I have assessed the statistics in my report.

Comments to author:

Summary: In this paper, the authors investigated the role of transposon cooption in generating novel genes. Specifically, they screened the human and mouse genomes and determined that the eutherian Bex/Tceal gene cluster is derived from two ancestral L1 elements. Using comparative genomics, they inferred that evolution of this locus required both L1 retrotransposon insertion into the locus and incorporation of a preexisting DNA sequence motif as a key regulatory element to generate the initial gene. This proto-Bex/Tceal gene was then subsequently expanded into a multi-gene cluster via gene duplication. The authors then investigated the possible biological function of the Bex/Tceal genes using a suite of assays including gene expression analyses, genetic knock-out of one gene, Bex3, and subsequent morphological, behavioral, and electrophysiological characterization of this mutant. In doing so, they determined that Bex/Tceal genes are expressed in mammalian brains, and that a subset of these genes are differentially expressed in autism and schizophrenia. They also found that Bex3-KO mice had subtle craniofacial differences and displayed altered behaviors relative to wild-type controls. Mutant mice also had fewer cortical and subcortical interneurons, and impaired excitatory inhibitory postsynaptic currents. Based on this data, the authors propose that the Bex3 gene plays a role in neural development and function.

General Comments: Overall, the paper is well written and presented, and describes an intriguing example of how new gene families can be born via transposon cooption. The authors integrate evolutionary and genomics analysis with a variety of genetic, molecular, and behavioral analysis, which strengthens the conclusions of the paper. The genetic evidence is especially bolstered by the use of two different Bex3 mutants. There are only a few issues that need to be addressed, along with some relatively minor edits (see Specific Comments).

While the majority of data analyses are sound, there are a few experiments that are less convincing. First, to reconstruct the Bex/Tceal protogene, the authors combine portions of the consensus sequence that correspond to what is seen for TCEAL7. Using this proxy gene, termed HALEX, the authors determine that the proliferative effects seen by overexpressing Bex3 and Tceal7 in chick embryos are not observed when HALEX expressed and interpret this to mean that the functions of Bex3/Tceal7 did not preexist in the ancestral gene. While this is plausible, it relies on the assumption that HALEX is a good proxy of the ancestral gene, which it may not be given the conventional methods of ancestral reconstruction rely on Bayesian or maximum-likelihood frameworks to accurately construct ancestral sequences. Additionally, it seems hard to interpret the results of these genes expressed in a non-eutherian, where these genes do not exist. Given these caveats, it is recommended that the authors deemphasize this point.

Second, the authors do not really describe the amplification of Bex/Tceal gene from the Tceal7 ancestor, nor provide any evolutionary analysis on natural selection working on genes within the family. There is a published report that describes gene conversion events that should be described and cited. The paper would be improved if the authors provided an alignment of the genes and describes how the genes

diverged and what parts of each gene are under selection. This would much better set up the experiments that only test the function of two family members (Bex3 and Tceal7).

Third, the authors emphasize that loss of Bex3 leads to craniofacial changes but these are very subtle effects. This is mitigated somewhat by the more dramatic behavioral and neuronal phenotypes, suggesting that Bex3 does indeed play a role in neuronal function. The authors should thus temper some of their conclusions when discussing the craniofacial data alone. Finally, based on their WB data of various mTOR targets from WT and Bex3 KO brain lysates, the authors make detailed claims about how the mTOR pathway could be the mediator of the physiological effects seen in Bex3 KO mice. However, the phosphorylation changes they observe in most cases are very subtle, and it is unclear whether this would really lead to enough dysregulation of the mTOR pathway to induce a phenotype. Thus, the authors should either place their results in the context of other known mTOR pathway effectors to further strengthen their point or perform a complementary assay that would supplement their initial results.

Specific Comments:

- * Abstract line 10, should be "these kind of events"
- * Page 3 line 37: "distinct" is misspelled
- * Page 3 line 42: While the example highlighted in this paper is interesting, transposon domestication is not a "unique" process but has been increasingly documented
- * Page 3 line 51: "Bex3 is essential for higher brain functions in placental mammals" is too strong a statement for the data as it is presented. Bex3 certainly appears to have an impact neuronal density and behavior in mice, but that does not necessarily mean it is essential or that it behaves the same way in other placental mammals
- * Page 7 line 10-20: Although the sequence of events that the authors hypothesize is plausible, it is difficult to be certain exactly which order the events occurred. The authors should acknowledge this possibility.
- * How similar are the Bex and Tceal genes? As a reader, I would find a phylogenetic tree and alignment of the genes/proteins to be useful. Are any of these genes or parts of these genes under purifying selection?
- * Page 7 line 37: Although expression of several Bex/Tceal genes are enriched in mammalian brains, this alone is a bit hard to interpret since many genes are more highly expressed in brain compared to other somatic tissues
- * Page 8 line 49: Figure 5E here should be Figure S5E
- * Page 9 line 37: The D24 measurement appears to not be marked on the figure schematic
- * Page 18 lines 20-27: Although it is certainly possible that the Bex3d24-72 allele is a dominant negative, the data from the craniofacial measurements, behavioral tests, and neurophysiology experiments generally suggests that this mutation produces less severe effects than the complete KO, which would be more in line with a hypomorph mutation
- * Page 21 lines 37-41: The authors should describe the algorithm or software package that they used to map the RNA-seq reads, as well as the parameters used. Additionally, it is not clear what the authors mean by "manually curated or introduced when necessary"
- * Fig S5, it would be useful to include the Bex proteins in part E, it's unclear why only the TCEAL proteins are shown. Also some discussion about which members might be functionally interchangeable, since only two (Bex3 and TCEAL7) were functionally tested in the chick electroporation experiment.
- * The paper would benefit from greater introduction/discussion on the known function/features of Bex/TCEAL proteins, including possible roles in cancer, gene conversion events, role in cell death, etc.

Reviewer 2

Are you able to assess all statistics in the manuscript, including the appropriateness of statistical tests used? Yes, and I have assessed the statistics in my report.

Comments to author:

Navas-Perez et al. present an interesting study that has two main components, i) characterisation of the Bex/Tceal gene cluster in placental mammals, including the discovery that Tceal7, including its ORF, is largely comprised of ancestral L1 sequences, and ii) mouse Bex3 KO and phenotype experiments. Both of these components are well executed and the phenotyping was very extensive. These findings are potentially important as they further explain the origins of the Bex/Tceal gene cluster, and it appears that mutations in this cluster are associated with several human neurological disorders. I enjoyed considering the manuscript and I would predict readers would view it similarly.

I have two main reservations around this work and I would welcome being able to consider a response from the authors.

1) Tceal7 is shown convincingly as a descendant of ancient L1 insertions, yet the phenotyping shown is of Bex3 KO. Why was Tceal7 KO not phenotyped instead? What is the exact relationship between Bex3 and Tceal7? I take it the two genes are descended from the same gene (the synthetic HALEX gene described by the authors). The Bex3 ORF is not identified as L1-derived in the genome browser. This question is intended to clarify the rationale for pursuing Bex3 and I apologise in advance if I have missed this somehow.

2) I would question whether some of the statistical analyses have been performed in the most desirable way. I was not clear on whether multiple testing correction was performed when more than two means were compared in the one test. In some comparisons (e.g. Figure 4A "rearing" from open field testing) distributions that almost entirely overlap achieve significance, and in these cases the use of SEM instead of SD becomes more questionable. When referring to Table S2, the authors say they observe significant expression differences for Tceal/Bex cluster genes amongst patients and controls for ASD and schizophrenia, when most of the FDR corrected values in Table S2 aren't significant, and the ones that are <0.05 are only barely significant. In other places the authors refer to significant results and yet I would question whether this reflects a notable difference in absolute terms, e.g. Figure 3E has tiny absolute differences in morphology and very significant p values. Certainly in many of the figures the reported differences are striking, but in others the interpretation is not as clear.

Other comments:

3) The "custom bioinformatic" pipeline could be explained a bit more thoroughly. It's interesting (although perhaps serendipity) that Tceal7 is only just barely over the txCdsPredict score threshold of 800, an explanation of why that was selected could be helpful.

4) It was good to see the many published examples of domesticated TEs listed in Table S1 used as positive controls for the bioinformatic pipeline. I noticed RAG1/RAG2 was missing from the list, and I am guessing that is because RepeatMasker doesn't annotate the ORFs of these genes as TE-derived because they are so ancient (PMID: 15898832). If that is the case it could be worth mentioning this general limitation in the text.

5) Gels and RT-PCR should be provided to show the Bex3 KOs were clean. The authors probably have these data already.

6) p17: the authors point out that mammalian-restricted genes have tissue-restricted expression patterns. It

could be worth pointing out that TEs also follow this trend in general (e.g. PMID: 19377475 or another ref the authors prefer) and so it is interesting that the TE-derived genes here are broadly expressed.

7) p6: "arouse from a mosaic" - typo "arose" and would suggest "composite" instead of "mosaic" due to the various meanings of the latter.

8) Bex is short for "brain-expressed X-linked". This cluster is known to be highly expressed in brain, and perhaps that should be stated at the outset on page 7.

GBIO-D-20-00598

Characterisation of a placental mammal-specific gene cluster generated by transposon domestication identifies *Bex3* as essential for advanced neurological functions

Enrique Navas-Pérez; Cristina Vicente-García; Serena Mirra; Demian Burguera; Noelia Fernández-Castillo; Jose Luis Ferrán; Macarena López Mayorga; Marta Alaiz-Noya; Irene Suarez-Pereira; Ester Antón-Galindo; Fausto Ulloa; Pol Cuscó; Rafael Falcón-Moya; Antonio Rodríguez-Moreno; Salvatore D’Aniello; Bru Cormand; Gemma Marfany; Eduardo Soriano; Ángel Carrión; Jaime J. Carvajal; Jordi Garcia-Fernandez.

Genome Biology

Response to reviewers

Dear Dr. Pang,

Thank you for sending the reviewers’ comments on the above paper, and for your indication of submitting a revised manuscript taking into account the referees’ concerns. We take the fact that both referees are broadly favorable and consider the work of potential interest. We detail below our responses to each reviewer. You will notice that we have included several additional analyses and figures which we trust enrich the manuscript, following the referees' suggestions, which we much appreciate.

Reviewer #1

General Comments: Overall, the paper is well written and presented, and describes an intriguing example of how new gene families can be born via transposon cooption. The authors integrate evolutionary and genomics analysis with a variety of genetic, molecular, and behavioral analysis, which strengthens the conclusions of the paper. The genetic evidence is especially bolstered by the use of two different *Bex3* mutants. There are only a few issues that need to be addressed, along with some relatively minor edits (see Specific Comments).

1) While the majority of data analyses are sound, there are a few experiments that are less convincing. First, to reconstruct the *Bex/Tceal* protogene, the authors combine portions of the consensus sequence that correspond to what is seen for TCEAL7. Using this proxy gene, termed HALEX, the authors determine that the proliferative effects seen by overexpressing *Bex3* and *Tceal7* in chick embryos are not observed when HALEX expressed and interpret this to mean that the functions of *Bex3/Tceal7* did not preexist in the ancestral gene. While this is plausible, it relies on the assumption that HALEX is a good proxy of the ancestral gene, which it may not be given the conventional methods of ancestral reconstruction rely on Bayesian or maximum-likelihood frameworks to accurately construct ancestral sequences. Additionally, it seems hard to interpret the results of these genes expressed in a non-eutherian, where these genes do not exist. Given these caveats, it is recommended that the authors deemphasize this point.

First, we thank the referee for his/her insightful comments on the manuscript. We acknowledge this is a good point and suggest a more appropriate name “ancestral protogene” for the *HALEX* element and not “ancestral gene”. We clarify in the manuscript that this protogene was reconstructed using the consensus sequences of HAL1b and L1MEe derived from RepeatMasker, not by employing the sequence of modern genes. Accordingly, we have expanded our explanation in the corresponding part of the *Methods* (page 31, lines 753-755) and *Results* section (pages 9-10, lines 210-229).

Being aware of the limitations of using chicken in our experiment, we tried to reconstruct the first steps of the evolutionary process (i.e. the *Bex/Tceal* protogene that originated from the insertion of a LIMEe copy into a HAL1b copy), not the very last common ancestor of the *Bex/Tceal* genes (which had already undergone its own evolutionary path). The rationale of expressing *HALEX*, *Bex3* and *Tceal7* in a non-eutherian organism (i.e. chicken) was to study the effects of these genes in a model that had not “co-evolved” with these genes, in order to mimic the possible response of the eutherian ancestor where the *HALEX* element first appeared. In this regard, we have further explained the rationale of this experiment in the corresponding part of the *Results* section (page 9, line 212-220).

We think that the fact that electroporation of both *Bex3* and *Tceal7*, but not *HALEX*, produce significant alterations in the proliferation ratio of the chicken neural tube (despite not being part of its gene networks), supports an scenario where the original *HALEX* element was not able to produce clear physiological modifications. This likely indicates that the *Bex/Tceal* genes progressively acquired their organismal functions integrating into pre-established molecular networks within the eutherian lineage. Nonetheless, we now acknowledge in the *Results* section that the cellular response observed in chicken might not reproduce the signalling and transcriptional environment of the placental ancestor. Moreover, the explanation concerning the interpretation of these results has been rewritten and clarified in the *Discussion* section (pages 15-16, lines 372-376).

2) Second, the authors do not really describe the amplification of *Bex/Tceal* gene from the *Tceal7* ancestor, nor provide any evolutionary analysis on natural selection working on genes within the family. There is a published report that describes gene conversion events that should be described and cited. The paper would be improved if the authors provided an alignment of the genes and describes how the genes diverged and what parts of each gene are under selection. This would much better set up the experiments that only test the function of two family members (*Bex3* and *Tceal7*).

We fully understand and acknowledge the point made by the reviewer and, for this reason, we have added a new section called “*Evolutionary diversification of the Bex/Tceal family*” in the *Results* (pages 6-7, lines 139-164).

To illustrate how this gene family evolved, we now include several new analyses. First, we show an alignment of the BEX and TCEAL human proteins together with a characterization of the structural features (coiled coil domains, α -helices and disordered regions) conserved among the gene family and the ancestral transposon (new Fig. S3A and Fig. S4; Fig. S4 was Fig. S5E in the previous version of the manuscript). Furthermore, we show a phylogenetic tree of the BEX/TCEAL proteins from several species representing major eutherian lineages (Fig. S4B), confirming that some groups of paralogues frequently homogenize their coding regions through gene conversion in all studied clades.

Second, with regard to positive selection, we guess the article indicated by the referee is “Adaptive evolution and frequent gene conversion in the brain expressed X-linked gene family in mammals”, by Liqing Zhang. In this article the author tried to analyse the selective pressures acting upon the *Bex* genes. This paper has now been referenced in the manuscript. Nonetheless, since then new and more accurate methods for identifying natural selection acting upon gene sequences have been developed. Thus, in order to characterise signatures of positive selection in the *Bex/Tceal* genes, we used the HyPhy suite. For this analysis we gathered the coding sequences of *Bex* and *Tceal* genes from eight eutherian species and excluded from the analysis the paralogs known to have suffered gene conversion between their coding regions. Given the strong sequence divergence between the genes of the *Bex* and *Tceal* subfamilies, we generated two sets containing: 1) only the *Bex* genes with *Tceal7* as an outgroup; 2) the *Tceal* genes with *Bex5* as an outgroup. The results of this analysis constitute a new set of data which we think improves the evolutionary side of the manuscript, as we now report signals of positive selection in some of the genes trough eutherian evolution. This data has been added to the *Results* section (page 7, lines 154-164) and is illustrated in Fig. S5.

3) Third, the authors emphasize that loss of Bex3 leads to craniofacial changes but these are very subtle effects. This is mitigated somewhat by the more dramatic behavioral and neuronal phenotypes, suggesting that Bex3 does indeed play a role in neuronal function. The authors should thus temper some of their conclusions when discussing the craniofacial data alone.

We agree that craniofacial changes are very subtle. Still, mutants and wild-type siblings could be visually distinguished in the lab, which is why we conducted the morphological analysis in the first place. To tune down this part, as suggested, we have moved most of the results of this analysis to Fig. S10, retaining only significant differences in Fig. 3, and adjusted the *Discussion* section to “*lead to subtle craniofacial changes and has a profound impact on repetitive and social behavioural performance, two important behaviour alterations required to diagnose autism spectrum disorders*”, in the line of the referee suggestion (pages 18-19, lines 446-448).

4) Finally, based on their WB data of various mTOR targets from WT and Bex3 KO brain lysates, the authors make detailed claims about how the mTOR pathway could be the mediator of the physiological effects seen in Bex3 KO mice. However, the phosphorylation changes they observe in most cases are very subtle, and it is unclear whether this would really lead to enough dysregulation of the mTOR pathway to induce a phenotype. Thus, the authors should either place their results in the context of other known mTOR pathway effectors to further strengthen their point or perform a complementary assay that would supplement their initial results.

We agree with the Reviewer that the differences in the mTOR pathway were modest among genotypes, and therefore it is unlikely that this pathway, alone, may be causative for the phenotype. Accordingly, we have tuned down the importance of the mTOR pathway at in the revised manuscript and suggested the participation of other pathways. (page 15, lines 354-364, and page 17, lines 407-411)

Specific Comments:

*Abstract line 10, should be "these kind of events"

We think that "The" relates to "Kind" and not to "Events", and thus should be singular. To facilitate reading we have changed "kind" to plural too (page 3, line 53).

*Page 3 line 37: "distinct" is misspelled

Corrected (page 3, line 65).

*Page 3 line 42: While the example highlighted in this paper is interesting, transposon domestication is not a "unique" process but has been increasingly documented

To tone down the sentence, we discarded this adjective (page 3, line 68).

*Page 3 line 51: "Bex3 is essential for higher brain functions in placental mammals" is too strong a statement for the data as it is presented. Bex3 certainly appears to have an impact neuronal density and behavior in mice, but that does not necessarily mean it is essential or that it behaves the same way in other placental mammals.

Following the referee suggestion, we have changed the term “essential” for the less strong “relevant” (page 3, line 71). As “essential” was also included in the original title of the article, we propose to change the title accordingly

*Page 7 line 10-20: Although the sequence of events that the authors hypothesize is plausible, it is difficult to be certain exactly which order the events occurred. The authors should acknowledge this possibility.

We agree with the referee. We have added an explanation to acknowledge this possibility (“*Although we cannot determine the precise order of the events leading to the assembly of the whole cluster*”) at the end of the corresponding *Results* section (page 8, lines 181-184).

*How similar are the Bex and Tceal genes? As a reader, I would find a phylogenetic tree and alignment of the genes/proteins to be useful. Are any of these genes or parts of these genes under purifying selection?

As indicated in our response to the referee’s main concern number 2), we have addressed these issues in a new section of *Results*, called “*Evolutionary diversification of the Bex/Tceal family*” (pages 6-7, lines 139-164 and Figs. S3-S5).

*Page 7 line 37: Although expression of several Bex/Tceal genes are enriched in mammalian brains, this alone is a bit hard to interpret since many genes are more highly expressed in brain compared to other somatic tissues.

We fully agree with the referee note, and we have changed the text to “*Although brain is an organ where many genes tend to be expressed [36], this result suggests that some of the neural functions reported in mouse and human for this gene family [11, 12, 37] might be conserved among the eutherian clade*” (page 8, lines 191-194). We apologise in advance in case we have missed the point made by the reviewer.

*Page 8 line 49: Figure 5E here should be Figure S5E.

Corrected (page 10, line 224). Also, former Figure S5 is now Figure S8.

*Page 9 line 37: The D24 measurement appears to not be marked on the figure schematic.

The D24 measurement appears now at the bottom of the dorsal view in Fig 3B.

*Page 18 lines 20-27: Although it is certainly possible that the Bex3d24-72 allele is a dominant negative, the data from the craniofacial measurements, behavioral tests, and neurophysiology experiments generally suggests that this mutation produces less severe effects than the complete KO, which would be more in line with a hypomorph mutation.

In line with the reviewer’s comment, we have removed the reference to the dominant negative and rewritten this part to make clear that the phenotypic differences between mutant lines suggest that *Bex3^{d(24-72)}* allele probably behaves as a hypomorph (“*In this regard, the less severe phenotype of Bex3^{d(24-72)} mice suggests that the resulting protein may act as a hypomorph version of the wild type allele*”) (page 19, lines 457-459).

*Page 21 lines 37-41: The authors should describe the algorithm or software package that they used to map the RNA-seq reads, as well as the parameters used. Additionally, it is not clear what the authors mean by “manually curated or introduced when necessary”.

We have added the information about the algorithm and mapping software (Bowtie) to the *Methods* section. The expression “manually curated” means that the transcript sequences of the index were improved (in terms of expanding sequence) when they were incomplete in the Ensembl transcripts database. “Introduced when necessary” means that some of the genes were absent in the Ensembl database and thus were introduced before creating the index. We added these clarifications to the *Methods* section (page 23, lines 549-554).

*Fig S5, it would be useful to include the Bex proteins in part E, it's unclear why only the TCEAL proteins are shown. Also some discussion about which members might be functionally interchangeable, since only two (Bex3 and TCEAL7) were functionally tested in the chick electroporation experiment.

We did not include BEX proteins in this figure of the submitted manuscript because a similar analysis had already been published in a paper by Fernández et al. 2015 (which we cite in the *Results* and *Discussion* sections; pages 6-7, lines 146-149 and page 10, line 244). However, we agree with the referee that it may be

better to include these data as well, hence now we include the secondary structure prediction for the BEX subfamily. We have divided figure S5 into two new supplementary figures, called figures S4 and S8. The aforementioned analysis is now in figure S4. These results are further discussed in page 16, lines 378-386.

We acknowledge that at least partially interchangeable roles are a very real possibility for some of the *Bex* and *Tceal* genes, especially for those with coding regions homogenized through gene conversion. However, from a gene family perspective, we think there is not enough knowledge on their functions to make a grounded statement. From one side, we decided to electroporate *Tceal7* because we know from our evolutionary analysis that it is the *Bex/Tceal* member most similar to the ancestral protogene. On the other hand, *Bex3* has a strong expression in murine embryonic neural tissues (see Figure 2B). Moreover, *Bex3* and *Tceal7* have been shown to regulate proliferation in several conditions in vitro, and thus were good candidates to produce a response in the tested organism [4-6, 8]. We have added an explanation regarding the choice of *Tceal7* and *Bex3* in the electroporation experiment (page 9, lines 217-220).

***The paper would benefit from greater introduction/discussion on the known function/features of Bex/TCEAL proteins, including possible roles in cancer, gene conversion events, role in cell death, etc.**

In line with this suggestion, we have included additional information summarizing previously known functions of *Bex/Tceal* genes in the *Background* (page 4, lines 88-90), *Results* (page 10, lines 232-237) and *Discussion* sections. Furthermore, we address the gene conversion events among *Bex/Tceal* genes in the “Evolutionary diversification of the *Bex/Tceal* family” section of *Results* (pages 6-7, lines 139-164).

Reviewer #2

I have two main reservations around this work and I would welcome being able to consider a response from the authors.

1) *Tceal7* is shown convincingly as a descendant of ancient L1 insertions, yet the phenotyping shown is of *Bex3* KO. Why was *Tceal7* KO not phenotyped instead? I take it the two genes are descended from the same gene (the synthetic HALEX gene described by the authors). The *Bex3* ORF is not identified as L1-derived in the genome browser. This question is intended to clarify the rationale for pursuing *Bex3* and I apologise in advance if I have missed this somehow.

We would like to thank the referee for his careful reading and revision of the manuscript. Regarding his first comment, we have included an evolutionary analysis of the *Bex/Tceal* gene family in order to make clear the relationship between the distant paralogs *Tceal7* and *Bex3* (see answer to point 2) of Referee #1 and the new “*Evolutionary diversification of the Bex/Tceal family*” section; pages 6-7, lines 139-164).

Moreover, at the beginning of the ‘*Generation of two independent Bex3 mutant lines*’ section, we have further explained why we chose *Bex3* as an interesting candidate to further understand the functional role of these genes (page 10, lines 232-237). We aimed to choose one gene of the family to generate a mutant and characterize it in depth. Reviewing literature on *in vitro* experiments, we found that *Bex3* is involved in neuronal physiology, and that it has been pinpointed as one of the main candidates to cause the neurological features of patients harboring deletions encompassing *BEX/TCEAL* genes. Furthermore, it shows neural-enriched expression in adult and embryonic tissues. On the other hand, *Tceal7* might have a particularly derived role (and thus probably not representative) in mice, as it shows strong expression in muscle rather than neural tissues. Because of our interest on muscle biology and gene regulation, our team is at present generating new KO alleles for *Tceal7*, but we are at the very early stages of this second project and applying for additional funding.

Finally, apart from *TCEAL7*, neither *BEX3* nor any other *BEX/TCEAL* gene is identified as L1-derived using our pipeline. This is because the sequences of these genes are too derived from the ancestral protogene to be detected by RepeatMasker as TE-derived. In mouse, a more fast-evolving species than human, not even *Tceal7* can be detected as L1-derived, while elephant and other slow-evolving mammals still present this similarity.

2) I would question whether some of the statistical analyses have been performed in the most desirable way. I was not clear on whether multiple testing correction was performed when more than two means were compared in the one test. In some comparisons (e.g. Figure 4A “rearing” from open field testing) distributions that almost entirely overlap achieve significance, and in these cases the use of SEM instead of SD becomes more questionable.

Statistical analyses were conducted as indicated in the corresponding *Methods* section (page 31, lines 766-774). When more than two means were compared in one test, we performed one-way ANOVA followed by Tukey’s Honest Significant Difference (HSD) tests, which enables the comparison between pairs of means with appropriate adjustment for the multiple testing. Specifically, and as indicated in the corresponding legends, the statistical analyses represented in Figures 3, 4, 5 (B-G), S10 and S11 were done this way. On the other hand, we used FDR multiple testing correction when analysing gene expression data from patients with autism spectrum disorder or schizophrenia (Table S2). We think the methodology used was appropriate and we apologise if it was not appropriately explained in the text.

Besides, we reckon the representation of some comparisons might be misleading, as in the mentioned “rearing” experiment. In this case, the group of wild-type animals displayed 28, 29, 32, 31, 27 and 35 rearing events, while *Bex3*^{KO} mice had 34, 37, 28, 37, 36 and 46. Thus, the calculated p-value for this comparison is 0.0479, barely significant, but still significant (one-way ANOVA, which can also be used to compare only

two means, after testing assumptions of normality and homogeneity of variances with Shapiro and Bartlett tests, respectively).

When representing the results of an experiment, the type of error bar used should be carefully considered. SD quantifies the variability among replicates, while SEM indicates the precision with which the population mean can be calculated with that experiment, that is, how far the obtained mean deviates from the true population mean. Given that we chose to plot as black dots the values of all replicates in all experiments in an attempt to inform the reader about experimental variability (which is what the SD would do), we decided to use SEM in the error bars to convey additional information.

When referring to Table S2, the authors say they observe significant expression differences for Tceal/Bex cluster genes amongst patients and controls for ASD and schizophrenia, when most of the FDR corrected values in Table S2 aren't significant, and the ones that are <0.05 are only barely significant.

Regarding altered expression of *BEX/TCEAL* genes in ASD/schizophrenia patients, our aim was to perform an exhaustive search in all available published articles and datasets, and in Table S2 we reported all the significant data found. We used $p\text{-value}<0.05$ as a criterion to report findings since FDR was not applied and $q\text{-value}$ was not available for some studies. In addition, the number of patients and conditions in each study were different. Even though for some studies no significant changes were found after multiple testing correction, around half of the reported findings show a $q\text{-value}<0.1$ ($\text{FDR}<10\%$), a threshold usually applied for significance in expression studies. We considered that reporting all this information was useful as we could describe a tendency to altered expression in all studies, half of them showing significance at $\text{FDR}<0.1$, and consistent direction on expression of all genes across studies (downregulation).

In other places the authors refer to significant results and yet I would question whether this reflects a notable difference in absolute terms, e.g. Figure 3E has tiny absolute differences in morphology and very significant p values. Certainly, in many of the figures the reported differences are striking, but in others the interpretation is not as clear.

Indeed, in some cases the reported differences between wild-type and mutant animals are not striking. However, the word “significant” was in our opinion scrupulously used whenever those differences were statistically significant, and thus, plausible from a statistical perspective. Whether the magnitude of the difference is small or large is a different matter. For example, Figure 3E reveals that cortical surface in *Bex3^{KO}* mice is significantly smaller than in wild-type and *Bex3^{A(24-72)}* animals (P values 0.0047 and 0.0247, respectively, one-way ANOVA with Tukey HSD test). The mean values for this parameter in wild-type, *Bex3^{KO}* and *Bex3^{A(24-72)}* mice are 32.65, 29.53 and 31.95 mm², respectively. Thus, cortical surface in *Bex3^{KO}* mice is roughly 10% smaller than in the other groups. Whether this small but significant difference is biologically relevant remains, certainly, to be elucidated.

Other comments:

3) The "custom bioinformatic" pipeline could be explained a bit more thoroughly. It's interesting (although perhaps serendipity) that Tceal7 is only just barely over the txCdsPredict score threshold of 800, an explanation of why that was selected could be helpful.

At the beginning of the *Results* section, we have added a brief description of the rationale behind the bioinformatic pipeline used to detect TE-derived genes (page 5, lines 102-104). As for the txCdsPredict score threshold, the UCSC Genome Browser states that “a score over 1000 is almost certain to be a protein, and scores over 800 have about a 90% chance of being a protein” (page 21, lines 509-512). Other authors have used the recommended 800 threshold to distinguish between coding and non-coding sequences (<https://www.ncbi.nlm.nih.gov/pmc/articles/PMC4702900/>).

4) It was good to see the many published examples of domesticated TEs listed in Table S1 used as positive controls for the bioinformatic pipeline. I noticed RAG1/RAG2 was missing from the list, and I am guessing that is because RepeatMasker doesn't annotate the ORFs of these genes as TE-derived because they are so ancient (PMID: 15898832). If that is the case it could be worth mentioning this general limitation in the text.

We appreciate the fine observation of the reviewer regarding the absence of RAG1/RAG2, and we also agree with his explanation. Ancient domestication events where the gene sequence is very derived from the TE source (or where no traces of the original element are found in the genome anymore) are virtually impossible to detect with a DNA-based method like ours. We have added this explanation to the text (page 5, lines 107-109).

5) Gels and RT-PCR should be provided to show the Bex3 KOs were clean. The authors probably have these data already.

We include the RT-PCRs of the transgenic lines following the referee indication (Additional file 1: Fig. S9, methods in page 25, lines 602-605). As indicated in the manuscript (page 10, lines 238-243) and depicted in figure 3A, the *Bex3*^{KO} line carries a 196-bp deletion starting at codon 13 that causes a frameshift mutation leading to the appearance of a premature STOP codon. The resulting peptide, if produced, would possibly be very unstable due to its short size (21 amino acids). Further, it would lack the pro-apoptotic and coiled-coil domains, which are responsible for Bex3 function. On the other hand, the *Bex3*^{Δ(24-72)} line, harbors a 147-bp mutation starting at codon 24 that removes 49 amino acids of the central core of the protein, leaving the coiled-coil domain intact. Since none of the mutations remove the promoter, the production of mRNA is not necessarily affected. Indeed, mRNA is produced, albeit shorter, in both lines and detected in the RT-PCR included in the revised version, in agreement with the deletions introduced.

Regarding antibodies, we tested three different commercially available anti-Bex3 (anti-NADE) antibodies in order to detect Bex3 protein and further corroborate the transgenic strains by western blot:

- A. NADE Polyclonal Antibody (TermoFisher Scientific, OSB00047W)
- B. NGFRAP1/BEX3/NADE Antibody (Novus Biologicals, NBP1-77149)
- C. NADE Antibody (TermoFisher Scientific, PA5-20076)

Mouse Bex3 protein comprises 124 amino acids, with a predicted molecular mass of 14,542 KDa (UniProt). However, according to the datasheets of all the tested commercial antibodies Bex3 migrates at ~23 kDa in SDS-PAGE, despite its predicted molecular weight. We followed the manufacturing instructions for SDS-PAGE experiments in both *Bex3* transgenic lines and control wild type mouse brain. Representative examples are presented below in raw western-blot images. Anti-NADE antibody by TermoFisher Scientific (OSB00047W) resulted in a multiple band pattern with a number of bands at significantly higher MW than expected (A). We observed a similar pattern by using anti- NGFRAP1/BEX3/NADE antibody by Novus Biologicals, observing a band at ~23 kDa as well (B). However, it was not the predominant signal. Similarly, we observed a band at ~25 kDa with a second anti-NADE antibody by TermoFisher Scientific (PA5-20076), but it is not the predominant band, again (C). For these reasons we considered all the tested commercial antibodies an unsuitable tool to corroborate our *Bex3* transgenic strains

Finally, we contacted the authors of the paper “NADE, a p75NTR-associated cell death executor, is involved in signal transduction mediated by the common neurotrophin receptor p75NTR” (Mukai J. et al., 2000), where an in-house antibody is used to detect murine Bex3. Unfortunately, they don’t have it anymore.

6) p17: the authors point out that mammalian-restricted genes have tissue-restricted expression patterns. It could be worth pointing out that TEs also follow this trend in general (e.g. PMID: 19377475 or another ref the authors prefer) and so it is interesting that the TE-derived genes here are broadly expressed.

We have added to the manuscript that TEs also tend to show tissue-restricted patterns, and cited the article suggested by the reviewer (page 18, lines 430-431).

7) p6: "arouse from a mosaic" - typo "arose" and would suggest "composite" instead of "mosaic" due to the various meanings of the latter.

We have corrected the typo and accepted the suggestion (page 6, line 124).

8) Bex is short for "brain-expressed X-linked". This cluster is known to be highly expressed in brain, and perhaps that should be stated at the outset on page 7.

We have added to the manuscript the origin of the acronyms for the *Bex* and *Tceal* families (page 6, lines 129-130).

Additional note: A new affiliation for Demian Burguera has been added. Also, we added an additional author, Carlos Herrera-Úbeda, who performed the analyses related to positive selection of the Bex/Tceal genes included in the new version of the manuscript.

Hoping that the revised version of the manuscript would be acceptable for publication in Genome Biology, on behalf of all the authors

Yours sincerely,

Jordi Garcia-Fernàndez

Second round of review

Reviewer 1

The authors have made a good faith effort to address all of our concerns.

Reviewer 2

The authors have carefully and thoughtfully addressed all of my comments.